# Better Models, Faster Training: Sigmoid Attention for single-cell Foundation Models

## Abstract

Training stable biological foundation models requires rethinking attention mechanisms: we find that using sigmoid attention as a drop in replacement for softmax attention a) produces better learned representations: on six diverse single-cell datasets, sigmoid achieves 25% higher cell-type separation, better cell-type cohesion metrics, and lower validation loss, b) faster training, models with sigmoid attention train up to 10% faster than their softmax counterparts, and c) more stable training by eliminating inherent sources of instability in softmax attention. We establish that sigmoid attention has globally bounded derivatives ($\leq 0.25$) as opposed to softmax, and a diagonal Jacobian structure in contrast with softmax's dense coupling, which together help alleviate training instabilities. In stress tests on 160M-parameter bidirectional attention models trained without gradient clipping on 8K-token sequences, softmax diverges catastrophically, with gradients exploding by four orders of magnitude, while sigmoid remains stable. Finally, we implement and open-source TRITONSIGMOID, an efficient GPU kernel that achieves 515 TFLOPS on H100 GPUs, outperforming both FlashAttention-2 and FlashSigmoid, with native padding support, which is essential for biological sequences. Our results establish sigmoid attention as both theoretically grounded and empirically superior for biological foundation models. Code will be made available upon publication.

## 1 Introduction

Foundation models trained on massive snapshots of the internet have increasingly become the dominant approach to modeling, understanding, and generating images and natural language (Devlin et al., 2019; Radford et al., 2019; Brown et al., 2020; Achiam et al., 2023; Meta, 2024). Similar foundation models are now trained on large-scale single-cell RNA sequencing (scRNA-seq) data, learning representations that capture cellular identity, state, and function (Chevalier et al., 2025; Theodoris et al., 2023; Cui et al., 2024; Schaar et al., 2024; Bian et al., 2024). These models enable zero-shot cell type annotation and discovery, perturbation prediction, and drug response modeling, with applications spanning developmental biology, pre-clinical target and biomarker discovery, and personalized medicine.

Although many modeling and architectural choices carry over from language, transcriptomic data poses distinctive challenges. Transformer-based single-cell foundation models treat the transcriptomic profile of a cell, the genes it expresses, as a sequence, and use the self-attention mechanism (Vaswani et al., 2017) to compute pairwise interactions between gene tokens, capturing co-expression patterns and regulatory relationships. Softmax attention, the de facto choice in language modeling, normalizes pairwise similarities to lie on the probability simplex. Thus, increasing attention on one gene token necessarily decreases attention on all others. However, a gene may be simultaneously and equally governed by multiple regulatory programs. For example, gene regulatory networks frequently exhibit co-regulation, where a single target gene is simultaneously activated by multiple transcription factors acting through independent enhancers (Spitz & Furlong, 2012; Alberts et al., 2002). A non-competitive attention mechanism that can attend strongly to many genes at once better models such data. Moreover, the normalization also contributes to numerical instabilities. As sequences grow longer and dot-product magnitudes increase, attention entropy collapses onto a small number of tokens, which can lead to catastrophic loss divergences Zhai et al. (2023); Hong & Lee

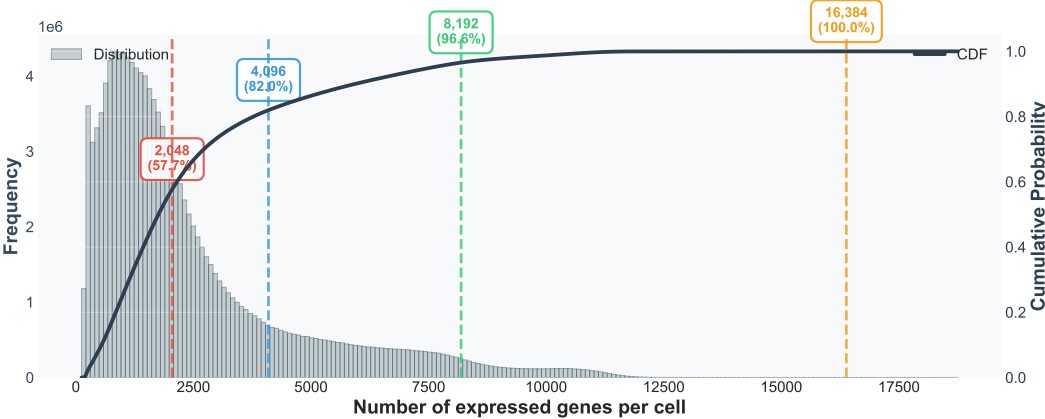

Figure 1: **Sequence length distribution in CellxGene pretraining dataset.** The histogram (left y-axis, gray bars) shows the distribution of sequence lengths across cells in the dataset, revealing high variability characteristic of biological data. The cumulative distribution function (right y-axis, black line) indicates the fraction of sequences below each length. Vertical dashed lines mark common context length thresholds (2,048, 4,096, 8,192, and 16,384 tokens) and percentage of cells fully covered at each. Unlike text data where multiple sequences can be packed together, biological sequences must be processed individually with padding, making padding-aware kernel implementations essential. The jagged distribution demonstrates why efficient padding support is critical for biological foundation models.

(2025). In single-cell foundation modeling (Chevalier et al., 2025; Cui et al., 2024; Theodoris et al., 2023), where transcriptomic profiles typically include thousands of genes, such instability is common, leading to failed runs and wasted computation. Recent work has explored sigmoid attention as an alternative in vision and language (Ramapuram et al., 2025; Qiu et al., 2025), replacing softmax with an element-wise sigmoid. While other element-wise attention mechanisms exist, we adopt sigmoid attention for training biological foundation models due to its relative maturity. Sigmoid attention provides theoretical stability guarantees (Section 3, Section A.1) and empirical performance advantages (Section 5), but practical adoption depends on computational efficiency. In biological foundation models, extreme variability in per-cell sequence length (Figure 1) makes padding-aware kernel implementations essential for practical deployment.

Transcriptomic data also pose unique computational challenges for the implementation of sigmoid attention. Depending on sequencing technology, the number of genes expressed in a cell ranges from a few hundred to several thousand. This *jaggedness* introduces critical issues. First, unlike text, where a paragraph can be split up into multiple short sequences to create *a fixed-length model input*, each cell represents an independent biological sample with its own masking pattern and cannot be split or concatenated without conflating distinct transcriptomic contexts. Each cell must therefore be processed individually, requiring padding to the maximum sequence length and resulting in significant computational waste on padded positions. Second, to prevent truncation of gene information and loss of biological signal, models must accommodate long context windows, which increases both memory demands and padding inefficiency. For example, Figure 1 plots the distribution of the number of genes expressed by a cell across CELLxGENE (CZI Single-Cell Biology Program et al., 2023), a collection of single-cell transcriptomic data amalgamated from diverse sources. Observe both the broad diversity in the number of genes expressed by a cell and that using a short context window (like 2048) would truncate genetic context for a substantial proportion (43%) of cells. Most existing foundation models such as scGPT (Cui et al., 2024) (1,200 tokens), Geneformer (Theodoris et al., 2023) (2,048 tokens), Tahoe-X1 (Gandhi et al., 2025) (2,048 tokens), and Transcriptformer (Pearce et al., 2025) (2,048 tokens) are limited to these shorter context windows. Current sigmoid attention implementations (Ramapuram et al., 2025) do not support padding, requiring identical sequence lengths within a batch, and are not compatible with modern GPUs such as NVIDIA Blackwell architectures, which are essential for large-context-length models.

**Our approach.** We investigate sigmoid attention for biological foundation models and make three contributions:

1. **Efficient implementation** (Section 4): We develop TritonSigmoid, a high-performance GPU kernel with native padding support that achieves 515 TFLOPS on NVIDIA H100 GPUs—exceeding FlashAttention-2 (361 TFLOPS) and FlashSigmoid (440 TFLOPS). By efficiently handling extreme jaggedness, TritonSigmoid makes long-context biological foundation models practical.

2. **Empirical performance** (Section 5): We train biological foundation models on the CellxGene dataset with both attention mechanisms under standard training conditions. Sigmoid attention achieves lower validation loss across all six held-out datasets, 25% higher cell-type separation (MMD), and superior cell-type cohesion, demonstrating better learned representations.

3. **Training stability** (Section 5.3): We demonstrate that sigmoid attention prevents catastrophic training failure under stress-test conditions, consistent with its theoretical stability properties (bounded derivatives, diagonal Jacobian). In experiment on 160M parameter models trained without gradient clipping on 8K-token sequences, softmax diverges with four-order-of-magnitude gradient explosions, while sigmoid remains stable throughout training.

Our results demonstrate that sigmoid attention is empirically superior for biological foundation models. The combination of better learned representations, efficient padding-aware implementation, and improved training stability positions sigmoid attention as a practical replacement for softmax in biological applications. We release our implementation as open-source software.

## 2 Related Work

**Biological Foundation Models** Recent foundation models for single-cell biology (Theodoris et al., 2023; Cui et al., 2024; Schaar et al., 2024; Bian et al., 2024; Chevalier et al., 2025) represent cells as sequences of gene expression tokens and train using masked language modeling on millions of cells. These models learn rich representations that capture cell type identity, developmental trajectories, and perturbation responses, enabling applications in cell type annotation, drug discovery, and disease modeling.

Most biological foundation models use softmax attention, despite biological data presenting unique challenges that can exacerbate softmax's known instabilities. Single-cell inputs are highly heterogeneous and sparse, with heavy-tailed expression distributions and substantial technical noise (e.g., dropout and library-size variation), which can lead to saturated attention weights and sensitivity to outliers. In addition, the long context windows needed to avoid truncating biological information (4K–16K tokens) further amplify numerical issues during training. Finally, gene regulation is modular and overlapping, suggesting benefits from non-competitive attention mechanisms that allow strong interactions with multiple tokens simultaneously, rather than forcing attention mass to concentrate on a small subset.

We provide the first systematic investigation of alternative attention mechanisms for biological foundation models, demonstrating that sigmoid attention offers superior training stability and better learned representations for this domain.

**Training Stability** Transformer training instability is well-documented (Li et al., 2022; Liu et al., 2020), with softmax attention identified as a key source (Dasoulas et al., 2021; Castin et al., 2024). Previous work shows that the softmax Lipschitz constant grows exponentially with attention scores (Kim et al., 2021), causing gradient explosion in deep networks. Standard mitigation strategies include gradient clipping (Pascanu et al., 2013), careful learning rate scheduling, and architectural modifications like pre-normalization (Xiong et al., 2020). However, these techniques add hyperparameter complexity and may not fully eliminate instability in challenging settings (e.g., long sequences, high learning rates).

Alternative attention mechanisms that avoid softmax's sensitivity could offer a more direct solution. Sigmoid attention replaces normalization with an element-wise nonlinearity, yielding uniformly bounded derivatives and a diagonal Jacobian structure. These properties eliminate cross-token gradient coupling and prevent

sensitivity from increasing with attention magnitude. We formalize this distinction through Jacobian and Lipschitz analysis (Section A.1) and demonstrate empirically that sigmoid attention prevents catastrophic training failures that occur with softmax under stress-test conditions (Section 5.3).

**Attention Mechanisms**   Self-attention (Vaswani et al., 2017) builds contextualized token representations by computing pairwise interactions between queries and keys. Although softmax attention is widely successful, it is known to exhibit numerical instabilities (Li et al., 2022; Hong & Lee, 2025), incurs $O(n^2)$ time and memory in sequence length, and can be sensitive to outlier logits. A range of alternatives have been proposed, including linear attention (Katharopoulos et al., 2020), which reduces complexity at the cost of expressiveness; sparse attention patterns (Child et al., 2019; Beltagy et al., 2020), which lower compute but depend on hand-designed or task-dependent sparsity; and gated mechanisms (Yang et al., 2024; Qiu et al., 2025), which modulate attention scores or weights to improve stability (Qiu et al., 2025).

Sigmoid attention (Ramapuram et al., 2025) replaces the softmax normalization with an element-wise sigmoid. Ramapuram et al. conducted a theoretical and empirical analysis of this mechanism and report improvements on standard NLP and vision benchmarks, but its behavior in biological foundation models has not been systematically studied. Single-cell inputs pose additional practical challenges, in particular extreme sequence-length variability (hundreds to thousands of gene tokens per cell), which makes efficient padded batching essential. achieving lower validation loss, better learned representations, and superior training stability (Section 5).

**Efficient Attention Implementations**   Standard attention implementations materialize the full $n \times n$ attention matrix, requiring $O(n^2)$ memory. FlashAttention (Dao et al., 2022; Dao, 2023) addresses this through tiling and SRAM optimization, achieving near-optimal performance for softmax attention. Other efficient implementations include xFormers (Lefaudeux et al., 2022), which provides block-sparse patterns and fused kernels with broad hardware support; PyTorch's native memory-efficient attention (PyTorch Team, 2023), which offers hardware-optimized softmax kernels integrated into PyTorch framework; and FlexAttention (Dong et al., 2024), which offers flexible masking and score modification through user-defined functions while maintaining memory efficiency.

However, these implementations are specifically designed for softmax attention and cannot be easily adapted to alternative attention functions. FlashSigmoid (Ramapuram et al., 2025) adapts FlashAttention for sigmoid attention but critically lacks padding support (requiring identical sequence lengths within a batch) and compatibility with modern GPU architectures (e.g., NVIDIA B200). Standard PyTorch implementations of sigmoid attention support padding but achieve only 41 TFLOPS forward and 91 TFLOPS backward on H100 GPUs—far below hardware capabilities. This creates a practical barrier: biological sequences require padding support, sigmoid attention offers stability and quality advantages, but no existing kernel handles both efficiently.

Our TRITONSIGMOID  kernel addresses these limitations through block-sparse computation with early exit on fully padded blocks, tiled execution with attention recomputation in backward pass, providing the first implementation that combines sigmoid attention with efficient padding support. We implement our kernel using Triton (Tillet et al., 2019), which integrates directly into the PyTorch framework.

## 3   Background on Sigmoid Attention

**Softmax Attention.**   Consider a single-head self-attention layer applied to an input sequence $X = (x_1, \ldots, x_n) \in \mathbb{R}^{n \times d}$. Standard softmax attention computes:

$$\text{Att}(X) = \text{softmax}\Big(\frac{QK^\top}{\sqrt{d}}\Big)V, \tag{1}$$

where $Q = W_Q X$, $K = W_K X$, $V = W_V X$ are learned linear projections, $\text{softmax}(\mathbf{a})_j = \frac{e^{a_i}}{\sum_j e^{a_{ij}}}$, for a vector $\mathbf{a} = [a_1, \ldots, a_D]^T$, and softmax is applied row-wise such that the attention weights for each query token live on the probability simplex.

**Sigmoid Attention.** Sigmoid attention replaces softmax normalization with an element-wise sigmoid:

$$\text{SigmoidAttn}(X) = \sigma\Big(\frac{QK^\top}{\sqrt{d}} + b\Big)V, \tag{2}$$

where $\sigma(a) = \frac{1}{1+e^{-a}}$ is applied element-wise and $b$ is a fixed or learnable bias term. Unlike softmax, sigmoid does not normalize across tokens—each attention weight $\text{SigmoidAttn}(X)_{ij} \in (0,1)$ is computed independently. Following Ramapuram et al. (2025), we set $b = -\log(n)$ where $n$ is the sequence length, which empirically approximates softmax normalization while maintaining the stability benefits of element-wise computation.

The fundamental distinction lies in how attention weights interact. In softmax attention, each weight $\text{Att}(X)_{ij}$ depends on all scores in the row through the normalization in the denominator of the softmax function. Increasing the attention on one token necessarily decreases all others, a competitive mechanism. In sigmoid attention, $\text{SigmoidAttn}(X)_{ij}$ depends only on the single dot product $q_i^\top k_j$. Attention weights are decoupled, allowing a query to potentially attend strongly to multiple keys simultaneously. Beyond providing a more realistic model for gene regulation, this decoupling also has consequences for training stability. Prior work (Dasoulas et al., 2021; Castin et al., 2024) shows that softmax's local Lipschitz constant grows exponentially with score magnitude, and its dense Jacobian couples all token interactions. When scores grow large or attention entropy collapses, gradients can explode. Sigmoid attention avoids these issues, the element-wise nonlinearity produces diagonal Jacobians with globally bounded derivatives. The maximum derivative of $\sigma(x)(1 - \sigma(x))$ is $1/4$, independent of input magnitude. For completeness, we synthesize these existing theoretical results (Dasoulas et al., 2021; Castin et al., 2024; Ramapuram et al., 2025) which establish favorable Lipschitz properties of the sigmoid attention and its Jacobian in Section A.1. We demonstrate the practical implications of this stability in Section 5.3 through stress-test experiment.

## 4 An Efficient Sigmoid Attention Kernel

We developed TRITONSIGMOID, an efficient Triton (Tillet et al., 2019) based implementation of the sigmoid attention kernel with native padding support. TRITONSIGMOID builds on the tiling and blocking strategies from FlashAttention-2 (Dao et al., 2022) and FlashSigmoid (Ramapuram et al., 2025), while introducing key innovations for padding support and performance optimization. We summarize these below and direct the reader to Section A.4.2 for details.

*Block-sparse computation.* We identify and skip entirely padded blocks on both query and key sides, avoiding wasted computation on masked regions.

*Fused operations.* Following FlashAttention's design, we fuse the attention computation into a single kernel, eliminating intermediate materialization of the attention matrix. For sigmoid attention, we use a hardware-optimized tanh-based approximation $\sigma(x) \approx 0.5(\tanh(x/2) + 1)$, leveraging fast tanh primitives available on modern GPUs.

*Backward pass decomposition.* We split the backward pass into two kernels: one computing $\frac{\partial L}{\partial Q}$ and another computing $\frac{\partial L}{\partial K}$ and $\frac{\partial L}{\partial V}$. This decomposition eliminates atomic operations in favor of direct gradient accumulation, improving both performance and numerical stability. In addition, we recompute sigmoid activation in each backward pass to keep memory efficiency.

*Memory access optimization.* We employ transposed memory reads to maximize memory throughput for transposed arrays such as $K^\top$, which is required for computing $QK^\top$.

**Implementation details.** Choosing to implement TRITONSIGMOID in Triton buys us future-proof design. Triton kernels are just-in-time (JIT) compiled to GPU-specific code, allowing the compiler to automatically re-target and optimize the same source for new architectures and to take advantage of improved compiler passes and hardware features without manual rewrites. We also perform autotuning on a carefully curated set of target GPU specific configurations. Our tuning strategy balances exploration breadth with tuning time, achieving near-optimal performance while completing in seconds or minutes. Finally, our implementation is

fully compatible with PyTorch's `torch.compile`, enabling straightforward integration into existing training pipelines with automatic fusion of surrounding operations.

## 4.1 TritonSigmoid leads to faster training of foundation models

In this section, we benchmark TRITONSIGMOID both in isolation and by measuring training time in GPU hours for training runs of biological foundation models at different model sizes. All experiments are conducted on NVIDIA H100 80GB SXM5 GPUs using BF16 precision.

**Kernel Performance** We vary sequence length from 512 to 16,384 tokens, setting batch size such that the total number of tokens is 16,384 (e.g., batch size 32 for sequence length 512, batch size 1 for sequence length 16,384). We test head dimensions of 64 and 128, with hidden dimension 2048 (corresponding to 32 and 16 attention heads, respectively). For padded sequences, we test with 0% and 25% padding. Each configuration is run for 250 iterations with 100ms warmup to ensure stable measurements, and we report mean performance with 99% confidence intervals.

We evaluate TRITONSIGMOID against three baselines: standard PyTorch attention, FlashAttention2, and FlashSigmoid. Following the benchmarking methodology of FlashAttention (Dao et al., 2022), we calculate theoretical FLOPs and measure wall-clock time to compute achieved TFLOPS (tera floating-point operations per second). This metric captures both the theoretical computational work and the actual execution time, providing a hardware-utilization measure that accounts for memory bandwidth, kernel launch overhead, and other practical considerations. Complete FLOP formulas and derivations are provided in Section A.4.1.

*Performance without padding.* On unpadded sequences, TRITONSIGMOID achieves $7.15\times$ average forward-pass speedup (range: $4.86$–$9.50\times$) and $2.44\times$ backward-pass speedup (range: $1.74$–$3.29\times$) compared to standard PyTorch attention. These results surpass both FlashAttention2 ($5.22\times$ forward, $2.13\times$ backward) and FlashSigmoid ($6.49\times$ forward, $2.24\times$ backward), where all speedups are relative to PyTorch. The performance advantage over FlashAttention2 stems from sigmoid attention's computational simplicity: while softmax requires row-wise normalization with exponentials and division, sigmoid applies only element-wise operations, resulting in fewer instructions and better hardware utilization.

*Peak performance.* At 16K context length with head dimension 128 and no padding, TRITONSIGMOID achieves 515.6 TFLOPS forward and 373.5 TFLOPS backward—17% faster than FlashSigmoid (439.7/341.6 TFLOPS), 43% faster than FlashAttention2 (360.6/312.5 TFLOPS), and $5.6\times$ faster than standard PyTorch attention (92.8/204.8 TFLOPS).

*Performance with padding.* On sequences with 25% padding, TRITONSIGMOID demonstrates substantial advantages. The forward pass achieves $14.58\times$ average speedup over PyTorch (range: $9.49$–$21.13\times$), while the backward pass achieves $4.57\times$ speedup (range: $2.93$–$6.28\times$). Compared to FlashAttention2 on padded sequences, TRITONSIGMOID is 29% faster in the forward pass and 13% faster in the backward pass. Flash-Sigmoid cannot handle padded sequences and is excluded from this benchmark. This capability is critical for biological applications: variable-length transcriptomic sequences (200–16,000 genes per cell) require efficient padding support for practical batched training.

*Padding overhead.* Our block-sparse algorithm minimizes padding overhead. When comparing TRITONSIGMOID 's performance between 0% and 25% padding, TFLOPS decrease by only 9.3% in both forward pass (438.4 to 397.5 TFLOPS) and backward pass (316.1 to 286.6 TFLOPS). This modest overhead shows that our kernel efficiently skips padded blocks rather than computing on masked positions.

*Detailed comparisons.* Figure 2 shows TFLOPS across all configurations (head dimensions 64 and 128, padding 0% and 25%, forward and backward passes). TRITONSIGMOID consistently matches or exceeds flash variants. Supplementary Figure 8 provides detailed latency comparisons.

**Training Speed Performance** We measured end-to-end training performance across four model sizes (160M, 400M, 600M, and 1.4B parameters) at three context lengths (2K, 4K, and 8K tokens). Each configuration was run for 10,000 steps on 16 H100 GPUs with batch size 32 (2 samples per GPU), capturing throughput measurements every 200 steps. We report mean throughput (steps/second) with 95% confidence

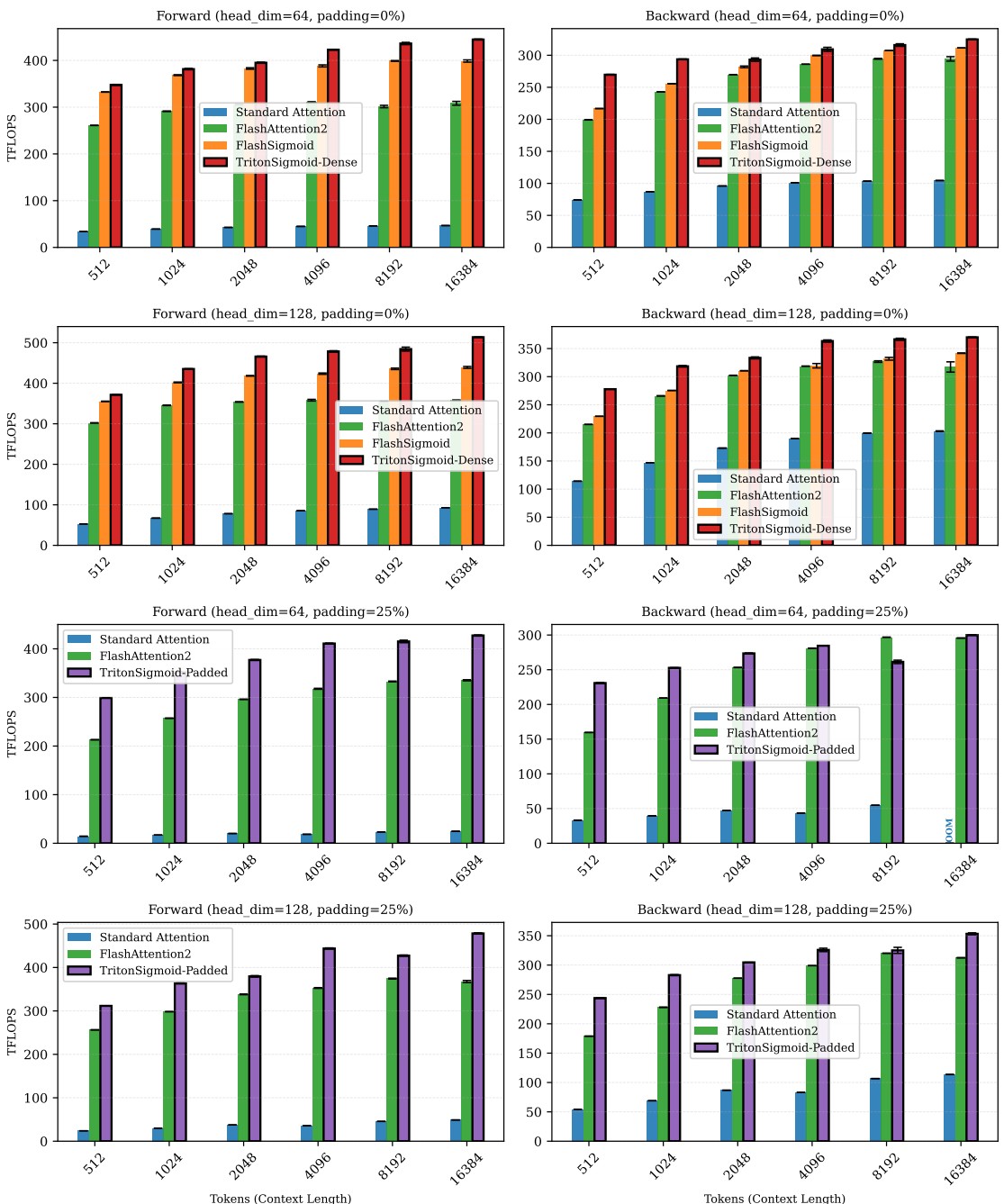

Figure 2: **TFLOPS comparison across attention implementations.** Performance across head dimensions (64, 128), padding levels (0%, 25%), and forward/backward passes. Rows 1–2: no padding, all implementations compared. Rows 3–4: 25% padding, FlashSigmoid unavailable (no padding support). Triton-Sigmoid matches or exceeds FlashSigmoid without padding and outperforms FlashAttention-2 across all configurations. Error bars: 99% confidence intervals. Context lengths: 512–16,384 tokens. H100 GPU, BF16 precision.

intervals across 50 measurements, then project these into GPU hours required to process 131.6M samples (the data used in our full training runs).

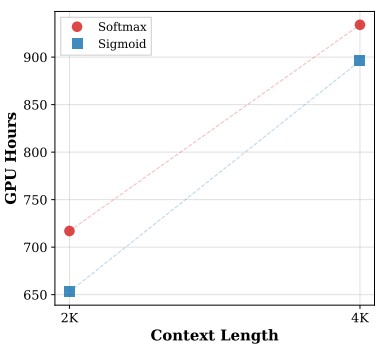 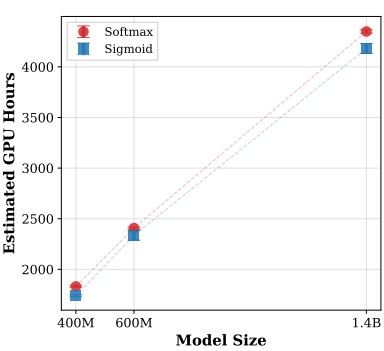 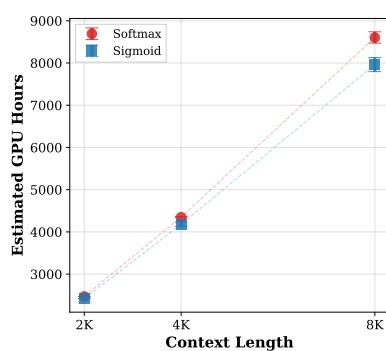

Figure 3: **End-to-end training compute cost.** Projected GPU hours to process 131.6M samples across four model sizes (160M, 400M, 600M, 1.4B) and three context lengths (2K, 4K, 8K), derived from measured throughput on 16 H100 GPUs (batch size 32; 2 samples/GPU). Sigmoid is consistently faster than softmax, with the gap increasing at longer contexts (e.g., 1.4B model: 2.1% at 2K, 4.0% at 4K, 7.5% at 8K), reflecting the quadratic scaling of attention with sequence length. Error bars show 95% confidence intervals.

As shown in Figure 3, sigmoid is consistently faster than softmax across all configurations. At 4K context, sigmoid achieves speedups across all model sizes: 400M (1739 vs 1832 GPU hours; 5.1% faster), 600M (2336 vs 2408; 3.0%), and 1.4B (4180 vs 4349; 4.0%). For the 1.4B model, the advantage grows with context length: 2.1% at 2K (2419 vs 2472 GPU hours), 4.0% at 4K, and 7.5% at 8K (7958 vs 8603 GPU hours), saving 645 GPU hours.

For the 160M model, we ran training to completion at both 2K and 4K context lengths. These wall-clock runs confirm the pattern: 653 vs 717 GPU hours at 2K (9% speedup) and 896 vs 934 GPU hours at 4K (4% speedup), consistent with the throughput-based projections (Figure 3).

Sigmoid's advantage grows with context length because attention cost scales quadratically with sequence length, making attention a larger fraction of total training compute at longer contexts. The end-to-end gains are smaller than the kernel-level FLOPS improvements (Section 4) because they include data loading, non-attention layers, and inter-GPU communication. Nevertheless, sigmoid provides consistent compute savings that scale with both model size and context length.

## 5 Sigmoid attention produces better single-cell foundation models

The kernel implementation (Section 4) demonstrates that TRITONSIGMOID achieves both higher computational throughput and faster end-to-end training. However, the critical question remains: do models trained with sigmoid attention produce better learned representations? We address this through comprehensive evaluation on single-cell biological foundation models trained to full convergence.

### 5.1 Experimental Setup

We train four 160M parameter models on the CellxGene dataset (CZI Single-Cell Biology Program et al., 2023) (131.6M cells, June 11, 2024 snapshot) under standard stable training conditions: 2K and 4K context lengths, each with softmax (FlexAttention) and sigmoid (TritonSigmoid) attention. We use FlexAttention for softmax because it integrates seamlessly with PyTorch's `torch.compile`, enabling end-to-end compilation and optimization. Our TRITONSIGMOID kernel is similarly designed for full `torch.compile` compatibility, ensuring both implementations benefit from automatic kernel fusion and optimization. All models use gradient clipping (threshold 1.0), learning rate $10^{-4}$, and train on full dataset to ensure full convergence. Full hyperparameters are listed in Section A.2.

We evaluate trained model performance using four complementary metrics on six diverse held-out single-cell datasets covering brain, blood, colon, lung, and heart tissue across developmental stages from embryonic to

aging (Section A.3.1). These datasets were not used during training and span different biological contexts (healthy, developmental, disease) to assess model generalization.

*Validation loss.* Measures masked language modeling performance via Monte Carlo estimation with multiple independent masking trials per cell. Lower loss indicates better generalization.

*SCIB biological conservation metrics.* Standard metrics from the scIB benchmarking framework (Luecken et al., 2022) quantifying whether embeddings preserve biological structure: cell-type silhouette (cohesion), Leiden NMI/ARI (clustering agreement), and isolated label scores (failure detection). Higher values indicate better structure preservation.

*UMAP visualization.* 2D projections of learned embeddings (via PCA then UMAP) colored by cell type, providing qualitative assessment of representation structure.

*Maximum Mean Discrepancy (MMD).* Quantifies separation between cell-type-specific representations in embedding space using an RBF kernel. Higher MMD indicates better distinguishability between cell types, which is desirable for downstream classification tasks.

Complete computational specifications for all metrics are in Section A.3.2.

## 5.2 Results

Figure 4 shows both training convergence and validation performance. During training (left panel), all four models converge smoothly over full dataset, with sigmoid and softmax reaching similar final loss values for both 2K and 4K contexts, with sigmoid 4K model achieving the lowest training loss. On held-out validation (right panels), six subplots show per-dataset performance. Each subplot displays four measurements: softmax and sigmoid at both 2K (circles) and 4K (squares) context lengths. Two clear patterns emerge: (1) Sigmoid (blue markers) consistently achieves lower validation loss than softmax (red markers) across all datasets and both context lengths, and (2) longer context (squares) outperforms shorter context (circles) for both attention mechanisms. Full loss values with 95% confidence intervals are in Table 5.

We next test whether sigmoid preserves biological structure in the learned embeddings. Using the scIB benchmarking framework (Luecken et al., 2022), we evaluate three metrics at 4K context: Leiden NMI (clustering agreement with ground-truth labels), Leiden ARI (pairwise clustering accuracy), and Silhouette label (cell-type cohesion). Figure 5 plots sigmoid versus softmax scores across six datasets. Sigmoid achieves the best cell-type cohesion on all six datasets. For clustering metrics, sigmoid performs better on 4 datasets. The aggregate score favors sigmoid on 4 of 6 datasets, suggesting sigmoid not only matches but slightly exceeds softmax at preserving biological structure.

To assess cell-type clustering and separation in the embedding space, we compute Maximum Mean Discrepancy (MMD) for all 28 pairwise cell-type comparisons in the Heart OFT dataset, using 1,000 bootstrap resamples to estimate each pairwise MMD. Sigmoid achieves a 25% higher mean bootstrap MMD on average, indicating stronger separation between cell types—a desirable property for downstream tasks that require distinguishing cell types. The full set of pairwise MMD values are in Table 1. For qualitative intuition, we also visualize the same embeddings using UMAP in Figure 7.

These results tell a clear story: sigmoid doesn't just match softmax—it often beats it. Validation loss favors sigmoid across all datasets at both 2K and 4K context lengths. The performance jump from 2K→4K in both mechanisms validates the importance of longer context windows for capturing complex gene relationships in biological foundation models. The embedding analysis reveals an additional advantage: sigmoid dominates cell-type cohesion on all six datasets and wins overall biological conservation on four of six. The MMD results are particularly notable—25% better cell-type separation despite UMAP visualizations that look similar.

Why does sigmoid achieve better embeddings with equivalent loss? One explanation lies in how attention is computed. Softmax normalizes across all positions, forcing a zero-sum competition: attending strongly to one gene means attending weakly to others. Sigmoid applies element-wise, allowing independent attention to multiple genes simultaneously without trade-offs. This flexibility may help capture the complex gene-gene relationships—co-expression patterns, regulatory networks, pathway memberships—that define cell-

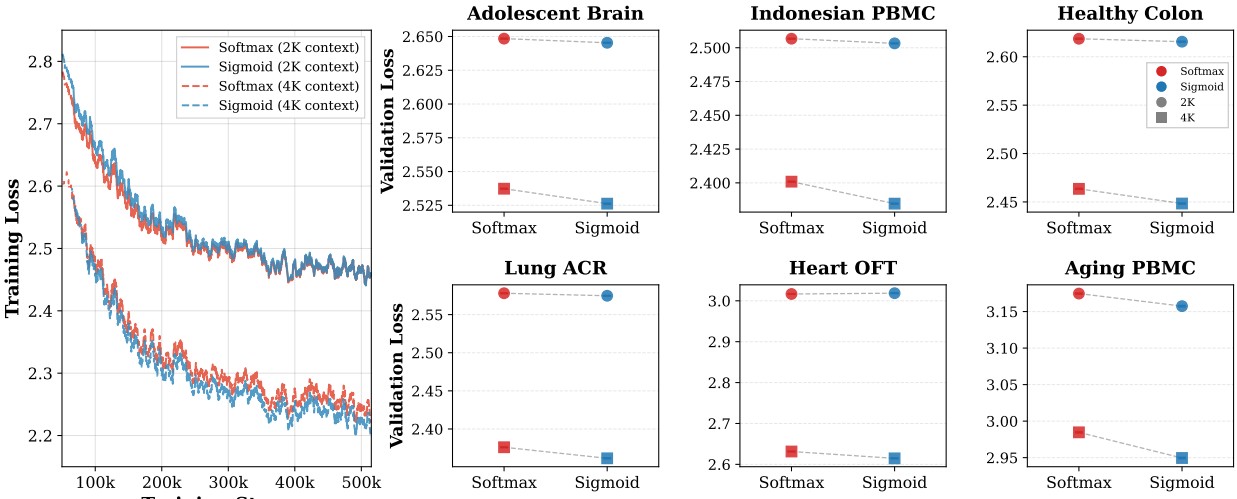

Figure 4: **Training convergence and validation performance across datasets. Left:** Training loss over full CellxGene dataset for four models (2K/4K context × sigmoid/softmax). All models converge smoothly with sigmoid 4K achieving the lowest final loss. **Right:** Six subplots showing validation loss per dataset. Each subplot displays four measurements: softmax (red) and sigmoid (blue) at 2K (circles) and 4K (squares) contexts. Two consistent patterns emerge: (1) Sigmoid systematically outperforms softmax across all datasets and both context lengths, and (2) longer context (4K) yields lower loss than shorter context (2K) for both attention mechanisms, demonstrating that extended context windows improve modeling of gene expression patterns.

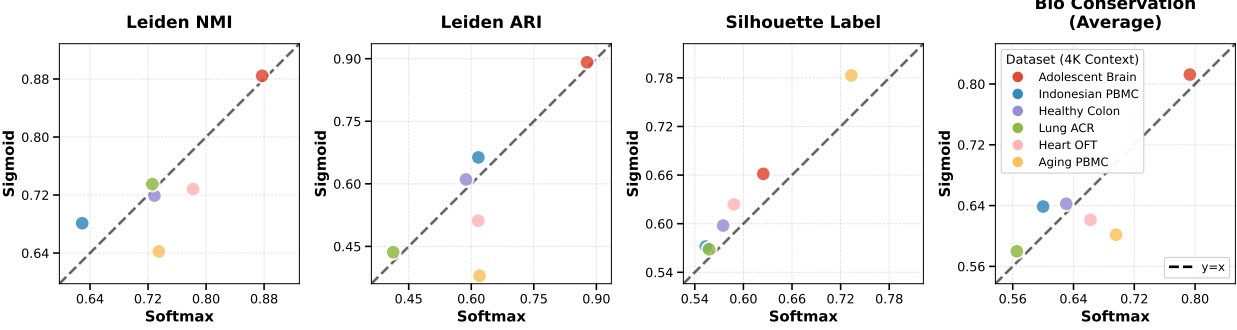

Figure 5: **SCIB biological conservation metrics at 4K context.** Scatter plots comparing sigmoid (y-axis) vs softmax (x-axis) across six datasets. Each panel shows a different metric: Leiden NMI (clustering agreement with ground-truth labels), Leiden ARI (pairwise clustering accuracy), Silhouette label (cell-type cohesion), and aggregate Bio Conservation (average of the three components). Points above the diagonal favor sigmoid. Sigmoid achieves perfect dominance on cell-type cohesion (6/6 datasets) and wins overall on 4/6 datasets for aggregate biological conservation, demonstrating superior preservation of biological structure compared to softmax.

type identity. The silhouette score analysis supports this: sigmoid better learns which genes distinguish cell types, even when both mechanisms achieve equivalent masked token prediction.

These empirical results demonstrate that sigmoid attention produces superior biological foundation models. Next, we examine sigmoid's stability advantages under stress-test conditions (Section 5.3).

Table 1: **Pairwise MMD for Heart OFT dataset at 4K context.** Each row shows one cell-type pair comparison. Values show MMD statistic with 95% bootstrap confidence interval [lower, upper] from 1000 bootstrap resamples. Best (highest) MMD per pair shown in bold. Sigmoid achieves higher MMD on all 28 pairs, with mean improvement of 25.0%.

| Cell Type Pair | Sigmoid MMD | Softmax MMD |
|---|---|---|
| Mesenchymal vs Cardiac | **0.467 [0.461, 0.472]** | 0.379 [0.375, 0.385] |
| Mesenchymal vs Endothelial | **0.316 [0.313, 0.321]** | 0.223 [0.220, 0.227] |
| Mesenchymal vs Immune | **0.381 [0.364, 0.399]** | 0.281 [0.273, 0.293] |
| Mesenchymal vs Neural | **0.366 [0.352, 0.382]** | 0.254 [0.243, 0.267] |
| Mesenchymal vs Adult endothelial | **0.385 [0.377, 0.399]** | 0.281 [0.277, 0.290] |
| Mesenchymal vs Adult valve interstitial | **0.545 [0.539, 0.551]** | 0.450 [0.446, 0.454] |
| Mesenchymal vs Adult immune | **0.525 [0.516, 0.537]** | 0.365 [0.359, 0.373] |
| Cardiac vs Endothelial | **0.528 [0.522, 0.534]** | 0.412 [0.408, 0.417] |
| Cardiac vs Immune | **0.443 [0.426, 0.462]** | 0.355 [0.344, 0.369] |
| Cardiac vs Neural | **0.491 [0.478, 0.507]** | 0.434 [0.422, 0.448] |
| Cardiac vs Adult endothelial | **0.459 [0.448, 0.474]** | 0.368 [0.360, 0.381] |
| Cardiac vs Adult valve interstitial | **0.676 [0.669, 0.683]** | 0.568 [0.562, 0.574] |
| Cardiac vs Adult immune | **0.628 [0.617, 0.642]** | 0.489 [0.480, 0.500] |
| Endothelial vs Immune | **0.435 [0.416, 0.455]** | 0.328 [0.317, 0.342] |
| Endothelial vs Neural | **0.459 [0.445, 0.475]** | 0.337 [0.326, 0.351] |
| Endothelial vs Adult endothelial | **0.427 [0.417, 0.442]** | 0.321 [0.314, 0.333] |
| Endothelial vs Adult valve interstitial | **0.655 [0.648, 0.662]** | 0.579 [0.574, 0.585] |
| Endothelial vs Adult immune | **0.638 [0.628, 0.649]** | 0.469 [0.462, 0.478] |
| Immune vs Neural | **0.666 [0.643, 0.694]** | 0.525 [0.509, 0.545] |
| Immune vs Adult endothelial | **0.634 [0.616, 0.656]** | 0.499 [0.485, 0.518] |
| Immune vs Adult valve interstitial | **0.460 [0.444, 0.478]** | 0.387 [0.376, 0.400] |
| Immune vs Adult immune | **0.627 [0.603, 0.654]** | 0.496 [0.478, 0.518] |
| Neural vs Adult endothelial | **0.760 [0.747, 0.779]** | 0.629 [0.619, 0.643] |
| Neural vs Adult valve interstitial | **0.571 [0.559, 0.584]** | 0.559 [0.549, 0.570] |
| Neural vs Adult immune | **0.881 [0.871, 0.895]** | 0.798 [0.785, 0.812] |
| Adult endothelial vs Adult valve interstitial | **0.339 [0.331, 0.352]** | 0.237 [0.231, 0.247] |
| Adult endothelial vs Adult immune | **0.578 [0.569, 0.592]** | 0.460 [0.453, 0.472] |
| Adult valve interstitial vs Adult immune | **0.615 [0.604, 0.628]** | 0.483 [0.474, 0.493] |

## 5.3 Training Stability: Stress Test

To validate the theoretical framework in Sections A.1 and 3, we design a stress test that isolates the attention mechanism's gradient dynamics. We train 160M parameter models for 80,000 training steps under deliberately harsh conditions: 8,192 token context, and *no gradient clipping*. All other hyperparameters are kept constant. The only variable is the attention mechanism: softmax versus sigmoid. Full experimental details are in Section A.2.

**Softmax Divergence** The softmax model initially learns successfully, with loss decreasing from $\sim$10 to $\sim$3 over the first 40,000 steps. However, between 40,000 to 60,000 steps, training becomes catastrophically unstable. Loss increases sharply to $\sim$10, while the global gradient norm explodes from $\sim$100 to $1.6 \times 10^6$—a four order-of-magnitude increase. Simultaneously, attention scores in layer 0 grow from $\sim$20 to 230 million. Training diverges permanently and cannot recover.

**Sigmoid Stability** We train an identical model with sigmoid attention and the results provide empirical validation: sigmoid attention completes all 80,000 training steps without divergence. Loss decreases monotonically from $\sim$10 to $\sim$3, maintaining smooth convergence throughout. Critically, around 55K steps—where softmax suffered catastrophic failure—sigmoid attention continues training stably with no anomalies. The gradient dynamics confirm theoretical predictions. While softmax gradient norms explode, sigmoid gradient norms remain bounded between 10–100 throughout all 80K steps, with a downward trajectory. Similarly, attention scores in sigmoid remain controlled, avoiding the exponential growth ($2 \times 10^8$) that destabilized

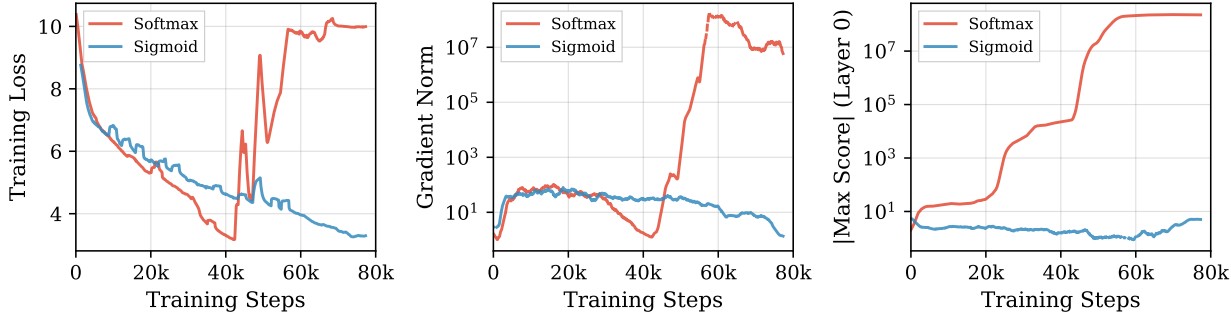

Figure 6: **Softmax divergence vs sigmoid stability under stress-test conditions.** Comparing training dynamics for identical 160M parameter models (8K context, no gradient clipping). **Left:** Training loss. Softmax initially learns (10→3 over 40K steps) but diverges catastrophically at step 55.6K (loss→10+). Sigmoid maintains stable monotonic decrease throughout 80K steps. **Middle:** Global gradient norm (log scale). Softmax explodes from ∼100 to $1.6 \times 10^6$ (four orders of magnitude) at divergence. Sigmoid remains bounded (10–100 range) throughout. **Right:** Absolute maximum attention score in layer 0 (log scale). Softmax scores grow exponentially to $2 \times 10^8$, while sigmoid scores remain controlled (1–5 range).

softmax. Figure 6 shows the complete comparison. Overall, this stress test isolates optimization stability at long context (8K+ tokens): softmax diverges, while sigmoid trains stably without gradient clipping.

# 6 Conclusion

Biological foundation models stress attention mechanisms in two ways: long contexts and extreme sequence-length variability across cells. To address both, we introduce TRITONSIGMOID, a padding-aware Triton kernel that makes sigmoid attention practical at scale, achieving 515 TFLOPS on H100 GPUs. With this implementation, we show empirically that sigmoid attention learns better representations than softmax across six single-cell datasets, with lower validation loss, 25% higher cell-type separation, and consistently stronger cell-type cohesion. We then provide theory explaining why sigmoid is more stable—its bounded derivatives and diagonal Jacobian remove key sources of softmax instability—and validate this with stress tests where softmax diverges catastrophically while sigmoid remains stable.

Together, these results establish sigmoid attention as a practical alternative to softmax for biological foundation models: efficient for jagged long-context sequences, better in representation quality, and more stable to train.

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

# A  Appendix

## A.1  Theoretical Foundations

In this section, we summarize previous results establishing why sigmoid attention leads to more stable training than softmax attention. We characterize the Lipschitz constants, Jacobian structure, and spectral norm bounds for both mechanisms.

### A.1.1  Lipschitz Structure and Gradient Coupling

The instability of softmax attention can be understood through the Jacobian of the attention operator with respect to its inputs. Prior work Dasoulas et al. (2021) shows that for any input matrix $X \in \mathbb{R}^{d \times n}$, the Frobenius norm of the derivative of the attention operator $D\mathrm{Att}_X$ satisfies:

$$\|D\mathrm{Att}_X\|_F \leq \|\mathrm{softmax}(S)\|_F + \sqrt{2}\|X^\top\|_{(\infty,2)} \|D(S)\|_{F,(2,\infty)}, \tag{3}$$

where $S = \frac{QK^\top}{\sqrt{d}}$ represents the scores (i.e. the query-key dot products). Equation 3 reveals that the instability of softmax attention arises from two distinct and interacting mechanisms: (i) a score magnitude term, controlled by the norm of the query-key dot products, and (ii) a probability mass concentration, governed by the Jacobian of the softmax normalization. Particularly, the Frobenius norm of $\mathrm{Att}_X$ decomposes into:

- **Score amplification term:** $\|D(S)\|$, which grows linearly with $\|Q\|$, and $\|K\|$,

- **Probability concentration term:** $\|\mathrm{softmax}(S)\|_F$, which measures the entropy of the distribution of attention weights.

**Minimum entropy.**  If the attention concentrates entirely on a single token (i.e. $A_{ik} = 1$ and $A_{ij} = 0$ for $j \neq k$), then $\|\mathrm{softmax}(S)\|_F = 1$. In order to understand the impact of this minimum entropy, we should consider that this is multiplied by the magnitude of the values $V$, and the input gradients. Also, subsequent works Kim et al. (2021); Castin et al. (2024) showed that the local Lipschitz constant of the softmax operator scales exponentially with the maximum score magnitude. More formally, the following proposition holds.

**Proposition 1.** *(Castin et al., 2024) Let $S = \frac{QK^\top}{\sqrt{d}} \in \mathbb{R}^{n \times n}$ denote the matrix of attention scores, namely the score function of input $X$. The local Lipschitz constant of the softmax operator satisfies:*

$$Lip_{loc}(\mathrm{softmax})(S) \leq C \exp(\|S\|_\infty), \tag{4}$$

*for a universal $C > 0$.*

These observations suggest that the instability of softmax attention is fundamentally tied to its normalization structure.

### A.1.2 Jacobian Structure and Decoupling

Examining the Jacobian matrices of softmax and sigmoid functions reveals the structural difference. For softmax,

$$\frac{\partial \text{softmax}(S)_{ij}}{\partial S_{ik}} = \text{softmax}(S)_{ij}\big(\delta_{jk} - \text{softmax}(S)_{ik}\big), \tag{5}$$

where $\delta$ is the Kronecker delta function. It is shown Castin et al. (2024) that this derivative is bounded by $O\Big(\exp\|X\|^2\Big)$. An important observation is that the derivative includes off-diagonal terms, making the Jacobian dense, and thus its spectral norm growing large when probability mass becomes concentrated.

On the other hand, for the sigmoid operator, the operations are element-wise, and the partial derivative is non-zero only when $j = k$:

$$\frac{\partial \sigma(S)_{ij}}{\partial S_{ik}} = \begin{cases} \sigma(S)_{ij}\big(1 - \sigma(S)_{ik}\big), & \text{if } j = k, \\ 0, & \text{if } j \neq k. \end{cases} \tag{6}$$

However, for the derivative function $\sigma'(x) = \frac{d\sigma}{dx} = \sigma(x)(1 - \sigma(x))$, we have:

$$\max_{x \in \mathbb{R}} \sigma'(x) = \sigma(0)(1 - \sigma(0)) = \frac{1}{4}. \tag{7}$$

Therefore, the local derivative of the sigmoid layer is independent of the input. Since the Jacobian is diagonal, its spectral norm equals its maximum diagonal entry. Consequently, the sigmoid attention operator is globally $1/4$-Lipschitz with respect to the score matrix.

### A.1.3 Spectral Norm Bound for Sigmoid Attention

Combining the bounded derivative with the linear projections yields the following result (see Theorem 3.2 (Ramapuram et al., 2025)):

**Theorem 1.** *Let $A = \{\langle W_Q x_i, W_K x_j\rangle\}_{ij}$ be the set of pre-activation scores. Then, the spectral norm of the Jacobian of sigmoid at input $X$ is bounded as follows:*

$$||J_{sigmoid}(X)||_2 \leq C \times \Big(\frac{1}{n}\sum_{i=1}^{n}||x_i||_2^2\Big). \tag{8}$$

*The constant $C$ depends only on $\|W_Q\|_2, \|W_K\|_2, \|W_V\|_2$, and the sigmoid bias scale, but is independent of sequence length and score magnitude.*

Theorem 1 establishes that sigmoid attention admits a uniform spectral norm bound on its Jacobian that is independent of attention sharpness and score magnitude. This follows from the element-wise sigmoid nonlinearity, which induces a diagonal Jacobian, and from the globally bounded derivative of the sigmoid function. As a result, sigmoid attention avoids the exponential sensitivity and cross-token gradient coupling inherent to softmax normalization, yielding a well-conditioned attention operator throughout training.

## A.2 Experimental Hyperparameters

This section provides complete hyperparameter specifications for all experiments reported in the paper. All experiments use the CellxGene dataset (CZI Single-Cell Biology Program et al., 2023) (131.6M cells, June 11, 2024 snapshot).

### A.2.1 Model Architecture

All models use a Transformer encoder architecture similar to Chevalier et al. (2025) with the following shared specifications in Table 2. The stress-test experiments designed to isolate attention mechanism stability use identical hyperparameters for both softmax and sigmoid variants, with the attention mechanism as the only variable. However we take away gradient clipping for the stress test experiment.

The performance comparison experiment trains four models (2 context lengths × 2 attention mechanisms) under standard stable training conditions to evaluate whether sigmoid attention matches softmax quality (see Table 3).

Table 2: **Model architecture specifications.** All experiments use 160M parameter models with these architectural choices.

| Parameter | Value |
|---|---|
| Model size | 160M parameters |
| Number of layers | 12 |
| Hidden dimension | 768 |
| Number of attention heads | 12 |
| Head dimension | 64 |
| FFN intermediate dimension | 3072 (4× hidden) |
| Activation function | GELU |
| Normalization | LayerNorm (pre-norm) |
| Dropout | 0.02 |
| Layer norm epsilon | $10^{-5}$ |
| Initializer range | 0.02 |
| Global batch size | 256 |
| Learning rate | $1 \times 10^{-4}$ |
| Weight decay | 0.1 |
| Optimizer | AdamW ($\beta_1 = 0.9, \beta_2 = 0.999, \epsilon = 10^{-8}$) |
| Learning rate schedule | Linear warmup (10K steps) + constant |
| Precision | BF16 (automatic mixed precision) |
| Gene masking probability | 0.15 |
| Random seed | 42 |

Table 3: **Hyperparameters for performance comparison.** Standard training conditions with gradient clipping to ensure stable optimization for both attention mechanisms.

| Parameter | 2K Context | 4K Context |
|---|---|---|
| Context length | 2,048 tokens | 4,096 tokens |
| Gradient clipping | 1.0 (L2 norm) | 1.0 (L2 norm) |
| Target duration | 515K steps | 515K steps |

### A.3 Evaluation Methodology

This section describes our evaluation protocol, including the datasets used for assessment and the metrics computed to quantify model performance.

### A.3.1 Evaluation Datasets

We evaluate model performance on six diverse single-cell datasets from CellxGene (CZI Single-Cell Biology Program et al., 2023), selected to cover a range of tissues, developmental stages, and disease contexts. These datasets were not used during training and provide independent assessment of model generalization. Table 4 summarizes key characteristics of each dataset.

All datasets are publicly available from CellxGene Collections (CZI Single-Cell Biology Program et al., 2023) at `https://cellxgene.cziscience.com/`. Original publications: Adolescent Brain (Galvao et al., 2026), Indonesian PBMC (Fachrul et al., 2026), Healthy Colon (Karakasheva et al., 2026), Aging PBMC (Gong et al., 2025), Lung ACR (Potter et al., 2026), Heart OFT (Leshem et al., 2025). CellxGene IDs: Adolescent Brain (09971952-46c4-410c-8fb1-0afa7b0b225f), Indonesian PBMC (45ccaf42-f198-4862-9f5f-2c4968071ff8), Healthy Colon (5cfcec6f-a400-4fed-a072-d3f27590132d), Aging PBMC (982c7fb5-a191-4125-

Table 4: **Evaluation datasets.** Six single-cell RNA-seq datasets from CellxGene covering diverse tissues, ages, and conditions.

| Short Name | Cells | Description |
|---|---|---|
| **Adolescent Brain** | 88,658 | Multimodal atlas of developing adolescent brain (ages 6-15) from cortex, hippocampus, and amygdala. Studies gene regulatory networks linking development to neuropsychiatric disease risk. |
| **Indonesian PBMC** | 71,492 | PBMCs from 199 Indonesians across Bali and New Guinea, capturing East Asian and Papuan ancestries in urban/rural and highland/lowland communities. Includes Neanderthal and Denisovan introgression signatures. |
| **Healthy Colon** | 19,059 | Endoscopic biopsies from 12 pediatric and 11 adult healthy controls. Epithelial and stromal fractions profiled separately. Contains 12 major cell types including specialized epithelial sub-clusters. |
| **Aging PBMC** | 9,354 | Cross-sectional cohort of 234 healthy adults (ages 40-89+) profiled to characterize age-related immune dynamics. Part of Immunobiology of Aging project. |
| **Lung ACR** | 39,200 | Lung transplant biopsies across acute cellular rejection (ACR), resolved ACR, and surveillance samples. Reveals persistent TGF-$\beta$ and mTOR signaling post-rejection. |
| **Heart OFT** | 30,125 | Developmental time series of human cardiac outflow tract: embryonic (CS16-17), fetal (week 12), and adult aortic valves. |

8ff5-edea84790468), Lung ACR (a5860cd7-61fc-433c-8452-343fd167e0e4), Heart OFT (e6c07fbd-c90b-48c0-b6e3-b03b2d7218c5).

### A.3.2 Evaluation Metrics

We provide detailed computational specifications for the evaluation metrics used in Section 5.

### A.3.3 Validation Loss

We compute validation loss via Monte Carlo estimation with multiple random masking patterns. For each cell, we perform 15 independent trials. In each trial, we randomly select 15% of the non-special gene tokens for masking, replace them with the `[MASK]` token, forward pass through the model, and compute cross-entropy loss over masked positions:

$$\ell_{i,t} = -\frac{1}{|M_{i,t}|} \sum_{j \in M_{i,t}} \log p(y_j \mid \mathbf{z}_{i,t}),$$

where $M_{i,t}$ is the set of masked positions in trial $t$ and $y_j$ is the true token. The final validation loss for cell $i$ is the mean across trials: $\bar{\ell}_i = \frac{1}{T} \sum_{t=1}^{T} \ell_{i,t}$, where $T = 15$. Using multiple masking patterns accounts for variance in which specific tokens are masked, providing robust estimates of model quality on the masked language modeling objective.

Table 5 provides complete validation loss results for all model configurations and evaluation datasets. The best (lowest) loss for each dataset is shown in bold. Results demonstrate that sigmoid achieves better performance than softmax under identical training conditions.

### A.3.4 UMAP Visualization

UMAP (Uniform Manifold Approximation and Projection) provides 2D visualization of learned embeddings to qualitatively compare representation structure between sigmoid and softmax attention. It is a nonlinear dimensionality reduction technique that preserves both local and global structure in high-dimensional data, making it well-suited for visualizing cell-type clustering patterns.

We extract embeddings $\mathbf{X} \in \mathbb{R}^{n \times d}$ from the model's learned representations, then apply PCA to capture 95% of the variance. This dimensionality reduction step improves UMAP's computational efficiency while

Table 5: **Detailed validation loss results.** Mean validation loss $\pm$ 95% confidence interval for all four model configurations across six held-out datasets. Each value is computed from 15 independent masking trials per cell. Best (lowest) loss per dataset is shown in bold. Sigmoid 4K model achieves best validation loss.

| Dataset | Sigmoid 2K | Softmax 2K | Sigmoid 4K | Softmax 4K |
|---|---|---|---|---|
| Adolescent Brain | $2.645 \pm 0.000$ | $2.648 \pm 0.000$ | $\mathbf{2.526 \pm 0.000}$ | $2.537 \pm 0.000$ |
| Aging PBMC | $3.157 \pm 0.001$ | $3.175 \pm 0.001$ | $\mathbf{2.950 \pm 0.001}$ | $2.985 \pm 0.001$ |
| Healthy Colon | $2.616 \pm 0.000$ | $2.619 \pm 0.000$ | $\mathbf{2.448 \pm 0.000}$ | $2.464 \pm 0.000$ |
| Heart OFT | $3.019 \pm 0.000$ | $3.017 \pm 0.000$ | $\mathbf{2.615 \pm 0.000}$ | $2.631 \pm 0.000$ |
| Indonesian PBMC | $2.503 \pm 0.000$ | $2.507 \pm 0.000$ | $\mathbf{2.385 \pm 0.000}$ | $2.401 \pm 0.000$ |
| Lung ACR | $2.575 \pm 0.000$ | $2.578 \pm 0.000$ | $\mathbf{2.362 \pm 0.000}$ | $2.376 \pm 0.000$ |

retaining the most informative directions. We compute UMAP projection with $n\_neighbors = 15$ (balancing local and global structure), $min\_dist = 0.5$ (allowing moderate point spacing).

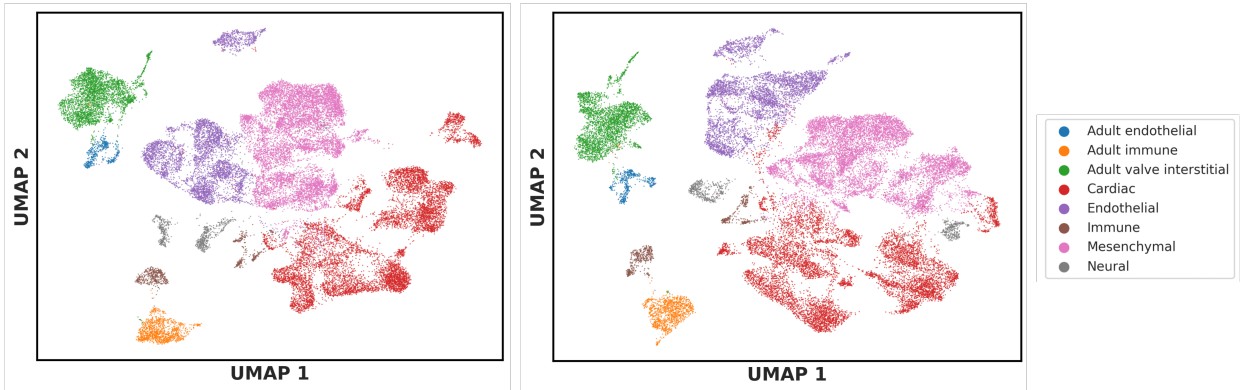

Figure 7: **UMAP visualization for Heart OFT dataset at 4K context.** Left: Softmax attention. Right: Sigmoid attention. We see that sigmoid model derived embeddings cluster better. For instance, endothelial cells (purple), are clustered closer by sigmoid, while softmax has two separate clusters.

### A.3.5  SCIB Biological Conservation Metrics

We evaluate biological structure preservation using standard metrics from the scIB benchmarking framework (Luecken et al., 2022). These metrics quantify whether learned embeddings capture biologically meaningful structure rather than technical artifacts. We compute three metrics: (1) **Silhouette label** measures cell-type cohesion (how tightly same-type cells cluster together), (2) **Leiden NMI** and (3) **Leiden ARI** measure clustering agreement (how well unsupervised clustering recovers ground-truth cell-type labels via information overlap and pairwise accuracy). Higher values indicate better biological structure preservation for all three metrics.

All metrics are computed using nearest neighbors ($k = 15$) in PCA-reduced embedding space (95% variance retained) following the standard scIB benchmarking protocol. Complete algorithmic details and metric definitions are available in the scIB metrics library (`https://github.com/YosefLab/scib-metrics`) and the original scIB paper (Luecken et al., 2022).

### A.3.6  Maximum Mean Discrepancy (MMD)

Maximum Mean Discrepancy (MMD) quantifies the separation between two distributions in embedding space. While SCIB metrics assess overall clustering quality, MMD directly measures how far apart different cell types are in the learned representation space. Higher MMD indicates greater separation between cell

types, which is desirable for downstream tasks that require distinguishing different biological states (e.g., differential expression analysis, cell-type classification).

For cell types $A$ and $B$ with embeddings $\mathbf{X}_A = \{x_1, \ldots, x_n\}$ and $\mathbf{X}_B = \{y_1, \ldots, y_m\}$, MMD is defined as:

$$\text{MMD}^2(\mathbf{X}_A, \mathbf{X}_B) = \mathbb{E}[k(x, x')] - 2\mathbb{E}[k(x, y)] + \mathbb{E}[k(y, y')] \tag{9}$$

where $k(\cdot, \cdot)$ is a kernel function and expectations are over pairs of points from each distribution. We use an RBF (Radial Basis Function) kernel with bandwidth selected via the median heuristic, which adapts to the local density of the embedding space. This formulation compares within-distribution similarity (first and third terms) against cross-distribution similarity (middle term).

We compute MMD for all pairwise cell-type comparisons in the Heart OFT dataset (28 pairs from 8 cell types). For each pair, we use bootstrap resampling (1000 iterations) to estimate uncertainty. Table 1 shows the complete pairwise MMD matrix for sigmoid vs softmax at 4K context. Sigmoid achieves higher MMD on all 28 pairs, with a mean improvement of 25.0%, indicating better separation of cell-type-specific representations.

## A.4 Additional Kernel Implementation Details

This section provides complete implementation details for the TritonSigmoid kernel, including FLOP calculations, algorithm pseudocode, and performance benchmarks.

### A.4.1 FLOP Calculation

We provide the complete formulas for computing theoretical FLOPs used in the TFLOPS benchmarking metric (Section 4).

For the forward pass with batch size $b$, number of heads $h$, sequence length $n$, and head dimension $d$, attention requires two matrix multiplications: $QK^\top$ (producing the attention scores) and the attention-weighted sum with $V$. The FLOPs for a single matrix multiplication are:

$$\text{FLOPs}_{\text{matmul}} = 2bhnd^2, \tag{10}$$

accounting for multiply-add operations. The total forward pass FLOPs are:

$$\text{FLOPs}_{\text{fwd}} = 2 \times \text{FLOPs}_{\text{matmul}} = 4bhn^2d. \tag{11}$$

For sequences with padding, we adjust $n$ to reflect only valid (non-padded) tokens. The backward pass requires $2.5\times$ the forward pass FLOPs, accounting for gradient computation ($2.0\times$) plus recomputation of activations ($0.5\times$):

$$\text{FLOPs}_{\text{bwd}} = 2.5 \times \text{FLOPs}_{\text{fwd}} = 10bhn^2d. \tag{12}$$

We compute TFLOPS as:

$$\text{TFLOPS} = \frac{\text{FLOPs}}{10^{12} \cdot \text{wall-clock time (seconds)}}. \tag{13}$$

### A.4.2 Algorithm Pseudocode

We provide detailed pseudocode for the TritonSigmoid kernel implementation. We present three algorithms: the forward pass (Algorithm 1), backward pass for query gradients (Algorithm 2), and backward pass for key/value gradients (Algorithm 3).

---

**Algorithm 1** TritonSigmoid Forward Pass with Padding Support

---

**Input:** Query $Q \in \mathbb{R}^{Z \times L_q \times H \times D}$, Key $K \in \mathbb{R}^{Z \times L_k \times H \times D}$, Value $V \in \mathbb{R}^{Z \times L_k \times H \times D}$
**Input:** Sequence lengths $n_q, n_k \in \mathbb{R}^Z$ (actual tokens per batch), bias $b \in \mathbb{R}^Z$
**Input:** Scale factor $\alpha$, block sizes $B_M, B_N$
**Output:** Output $O \in \mathbb{R}^{Z \times L_q \times H \times D}$

1: **Grid:** Launch $(\lceil L_q/B_M \rceil, Z \times H)$ thread blocks
2:
3: **for** each thread block $(m, zh)$ in parallel **do**
4:     $z \leftarrow \lfloor zh/H \rfloor$, $h \leftarrow zh \bmod H$                                             ▷ Batch and head indices
5:     $\text{start}_m \leftarrow m \cdot B_M$
6:     $\mathcal{M} \leftarrow \{\text{start}_m, \dots, \text{start}_m + B_M - 1\}$                        ▷ Query token indices
7:
8:     **if** $\text{start}_m \geq n_q[z]$ **then**                                     ▷ Skip fully padded blocks
9:         $O[z, \mathcal{M}, h, :] \leftarrow 0$
10:         **continue**
11:     **end if**
12:
13:     Load $Q_{\text{block}} \leftarrow Q[z, \mathcal{M}, h, :]$                                     ▷ $[B_M \times D]$
14:     Initialize $\text{Acc} \leftarrow \mathbf{0}_{B_M \times D}$ in FP32                            ▷ Accumulator
15:
16:     **for** $\text{start}_n = 0$ to $n_k[z]$ step $B_N$ **do**                ▷ Loop over key/value blocks
17:         $\mathcal{N} \leftarrow \{\text{start}_n, \dots, \text{start}_n + B_N - 1\}$                ▷ Key token indices
18:         Load $K_{\text{block}} \leftarrow K[z, \mathcal{N}, h, :]$                         ▷ $[B_N \times D]$
19:         Load $V_{\text{block}} \leftarrow V[z, \mathcal{N}, h, :]$                         ▷ $[B_N \times D]$
20:
21:         $S \leftarrow Q_{\text{block}} \cdot K_{\text{block}}^\top$                  ▷ Attention scores $[B_M \times B_N]$
22:         $S \leftarrow S \cdot \alpha + b[z]$                             ▷ Scale and bias
23:
24:                                       ▷ Mask padded positions before sigmoid
25:         $\text{mask}_m \leftarrow (\mathcal{M} < n_q[z])$, $\text{mask}_n \leftarrow (\mathcal{N} < n_k[z])$
26:         $S \leftarrow S + (-\infty) \cdot (1 - \text{mask}_m \otimes \text{mask}_n)$         ▷ Set padded to $-\infty$
27:
28:         $P \leftarrow \sigma(S)$                        ▷ Sigmoid activation (element-wise)
29:
30:         $\text{Acc} \leftarrow \text{Acc} + P \cdot V_{\text{block}}$               ▷ Accumulate weighted values
31:     **end for**
32:
33:     Store $O[z, \mathcal{M}, h, :] \leftarrow \text{Acc}$                         ▷ Write output block
34: **end for**

---

---

**Algorithm 2** TritonSigmoid Backward Pass: Query Gradients (dQ)

---

**Input:** Query $Q \in \mathbb{R}^{Z \times L_q \times H \times D}$, Key $K \in \mathbb{R}^{Z \times L_k \times H \times D}$, Value $V \in \mathbb{R}^{Z \times L_k \times H \times D}$
**Input:** Output gradient $\frac{\partial \mathcal{L}}{\partial O} \in \mathbb{R}^{Z \times L_q \times H \times D}$
**Input:** Sequence lengths $n_q, n_k \in \mathbb{R}^Z$, bias $b \in \mathbb{R}^Z$, scale $\alpha$, block sizes $B_M, B_N$
**Output:** Query gradient $\frac{\partial \mathcal{L}}{\partial Q} \in \mathbb{R}^{Z \times L_q \times H \times D}$

1:   **Grid:** Launch $(\lceil L_q / B_M \rceil, Z \times H)$ thread blocks
2:
3:   **for** each thread block $(m, zh)$ in parallel **do**
4:      $z \leftarrow \lfloor zh/H \rfloor$, $h \leftarrow zh \bmod H$            ▷ Batch and head indices
5:      $\text{start}_m \leftarrow m \cdot B_M$
6:      $\mathcal{M} \leftarrow \{\text{start}_m, \ldots, \text{start}_m + B_M - 1\}$            ▷ Query token indices
7:
8:      **if** $\text{start}_m \geq n_q[z]$ **then**            ▷ Skip fully padded blocks
9:         $\frac{\partial \mathcal{L}}{\partial Q}[z, \mathcal{M}, h, :] \leftarrow 0$
10:         **continue**
11:      **end if**
12:
13:      Load $Q_{\text{block}} \leftarrow Q[z, \mathcal{M}, h, :]$            ▷ $[B_M \times D]$
14:      Load $\frac{\partial \mathcal{L}}{\partial O_{\text{block}}} \leftarrow \frac{\partial \mathcal{L}}{\partial O}[z, \mathcal{M}, h, :]$            ▷ $[B_M \times D]$
15:      Initialize $\frac{\partial \mathcal{L}}{\partial Q_{\text{acc}}} \leftarrow \mathbf{0}_{B_M \times D}$ in FP32            ▷ Accumulator
16:
17:      **for** $\text{start}_n = 0$ to $n_k[z]$ step $B_N$ **do**            ▷ Loop over key/value blocks
18:         $\mathcal{N} \leftarrow \{\text{start}_n, \ldots, \text{start}_n + B_N - 1\}$            ▷ Key token indices
19:         Load $K_{\text{block}} \leftarrow K[z, \mathcal{N}, h, :]$            ▷ $[B_N \times D]$
20:         Load $V_{\text{block}} \leftarrow V[z, \mathcal{N}, h, :]$            ▷ $[B_N \times D]$
21:
22:                    ▷ Recompute forward pass (no materialization in forward)
23:         $S \leftarrow Q_{\text{block}} \cdot K_{\text{block}}^{\top}$            ▷ Attention scores $[B_M \times B_N]$
24:         $S \leftarrow S \cdot \alpha + b[z]$            ▷ Scale and bias
25:
26:                    ▷ Mask padded positions before sigmoid
27:         $\text{mask}_m \leftarrow (\mathcal{M} < n_q[z])$, $\text{mask}_n \leftarrow (\mathcal{N} < n_k[z])$
28:         $S \leftarrow S + (-\infty) \cdot (1 - \text{mask}_m \otimes \text{mask}_n)$            ▷ Set padded to $-\infty$
29:
30:         $P \leftarrow \sigma(S)$            ▷ Sigmoid activation
31:
32:                    ▷ Compute gradient through sigmoid and attention
33:         $\frac{\partial \mathcal{L}}{\partial O} V^{\top} \leftarrow \frac{\partial \mathcal{L}}{\partial O_{\text{block}}} \cdot V_{\text{block}}^{\top}$            ▷ $[B_M \times B_N]$
34:         $\frac{\partial \mathcal{L}}{\partial S} \leftarrow P \odot (1 - P) \odot (\frac{\partial \mathcal{L}}{\partial O} V^{\top})$            ▷ Sigmoid derivative
35:
36:                    ▷ Accumulate gradient w.r.t. Q
37:         $\frac{\partial \mathcal{L}}{\partial Q_{\text{acc}}} \leftarrow \frac{\partial \mathcal{L}}{\partial Q_{\text{acc}}} + \frac{\partial \mathcal{L}}{\partial S} \cdot K_{\text{block}}$            ▷ $[B_M \times D]$
38:      **end for**
39:
40:      $\frac{\partial \mathcal{L}}{\partial Q_{\text{acc}}} \leftarrow \frac{\partial \mathcal{L}}{\partial Q_{\text{acc}}} \cdot \alpha$            ▷ Apply scale factor
41:      Store $\frac{\partial \mathcal{L}}{\partial Q}[z, \mathcal{M}, h, :] \leftarrow \frac{\partial \mathcal{L}}{\partial Q_{\text{acc}}}$            ▷ Write gradient block
42: **end for**

---

---

**Algorithm 3** TritonSigmoid Backward Pass: Key/Value Gradients (dK, dV)

---

**Input:** Query $Q \in \mathbb{R}^{Z \times L_q \times H \times D}$, Key $K \in \mathbb{R}^{Z \times L_k \times H \times D}$, Value $V \in \mathbb{R}^{Z \times L_k \times H \times D}$
**Input:** Output gradient $\frac{\partial \mathcal{L}}{\partial O} \in \mathbb{R}^{Z \times L_q \times H \times D}$
**Input:** Sequence lengths $n_q, n_k \in \mathbb{R}^Z$, bias $b \in \mathbb{R}^Z$, scale $\alpha$, block sizes $B_M, B_N$
**Output:** Key gradient $\frac{\partial \mathcal{L}}{\partial K} \in \mathbb{R}^{Z \times L_k \times H \times D}$, Value gradient $\frac{\partial \mathcal{L}}{\partial V} \in \mathbb{R}^{Z \times L_k \times H \times D}$

1: **Grid:** Launch $(\lceil L_k/B_N \rceil, Z \times H)$ thread blocks
2:
3: **for** each thread block $(n, zh)$ in parallel **do**
4: $\quad z \leftarrow \lfloor zh/H \rfloor, h \leftarrow zh \bmod H$                  ▷ Batch and head indices
5: $\quad \text{start}_n \leftarrow n \cdot B_N$
6: $\quad \mathcal{N} \leftarrow \{\text{start}_n, \ldots, \text{start}_n + B_N - 1\}$           ▷ Key/value token indices
7:
8: $\quad$ **if** $\text{start}_n \geq n_k[z]$ **then**                ▷ Skip fully padded blocks
9: $\quad\quad \frac{\partial \mathcal{L}}{\partial K}[z, \mathcal{N}, h, :] \leftarrow 0, \frac{\partial \mathcal{L}}{\partial V}[z, \mathcal{N}, h, :] \leftarrow 0$
10: $\quad\quad$ **continue**
11: $\quad$ **end if**
12:
13: $\quad$ Load $K_{\text{block}} \leftarrow K[z, \mathcal{N}, h, :]$                     ▷ $[B_N \times D]$
14: $\quad$ Load $V_{\text{block}} \leftarrow V[z, \mathcal{N}, h, :]$                     ▷ $[B_N \times D]$
15: $\quad$ Initialize $\frac{\partial \mathcal{L}}{\partial K_{\text{acc}}} \leftarrow \mathbf{0}_{B_N \times D}$ in FP32        ▷ Accumulator for dK
16: $\quad$ Initialize $\frac{\partial \mathcal{L}}{\partial V_{\text{acc}}} \leftarrow \mathbf{0}_{B_N \times D}$ in FP32        ▷ Accumulator for dV
17:
18: $\quad$ **for** $\text{start}_m = 0$ to $n_q[z]$ step $B_M$ **do**        ▷ Loop over query blocks
19: $\quad\quad \mathcal{M} \leftarrow \{\text{start}_m, \ldots, \text{start}_m + B_M - 1\}$      ▷ Query token indices
20: $\quad\quad$ Load $Q_{\text{block}} \leftarrow Q[z, \mathcal{M}, h, :]$               ▷ $[B_M \times D]$
21: $\quad\quad$ Load $\frac{\partial \mathcal{L}}{\partial O_{\text{block}}} \leftarrow \frac{\partial \mathcal{L}}{\partial O}[z, \mathcal{M}, h, :]$         ▷ $[B_M \times D]$
22:
23:                       ▷ Recompute forward pass (transposed perspective)
24: $\quad\quad S^\top \leftarrow K_{\text{block}} \cdot Q_{\text{block}}^\top$              ▷ Attention scores $[B_N \times B_M]$
25: $\quad\quad S^\top \leftarrow S^\top \cdot \alpha + b[z]$                   ▷ Scale and bias
26:
27:                       ▷ Mask padded positions before sigmoid
28: $\quad\quad \text{mask}_m \leftarrow (\mathcal{M} < n_q[z]), \text{mask}_n \leftarrow (\mathcal{N} < n_k[z])$
29: $\quad\quad S^\top \leftarrow S^\top + (-\infty) \cdot (1 - \text{mask}_n \otimes \text{mask}_m)$     ▷ Set padded to $-\infty$
30:
31: $\quad\quad P^\top \leftarrow \sigma(S^\top)$                     ▷ Sigmoid activation
32:
33:                       ▷ Accumulate gradient w.r.t. V
34: $\quad\quad \frac{\partial \mathcal{L}}{\partial V_{\text{acc}}} \leftarrow \frac{\partial \mathcal{L}}{\partial V_{\text{acc}}} + P^\top \cdot \frac{\partial \mathcal{L}}{\partial O_{\text{block}}}$         ▷ $[B_N \times D]$
35:
36:                       ▷ Compute gradient through sigmoid and attention for K
37: $\quad\quad V(\frac{\partial \mathcal{L}}{\partial O})^\top \leftarrow V_{\text{block}} \cdot \frac{\partial \mathcal{L}}{\partial O_{\text{block}}}^\top$        ▷ $[B_N \times B_M]$
38: $\quad\quad \frac{\partial \mathcal{L}}{\partial S^\top} \leftarrow P^\top \odot (1 - P^\top) \odot (V(\frac{\partial \mathcal{L}}{\partial O})^\top)$    ▷ Sigmoid derivative
39:
40:                       ▷ Accumulate gradient w.r.t. K
41: $\quad\quad \frac{\partial \mathcal{L}}{\partial K_{\text{acc}}} \leftarrow \frac{\partial \mathcal{L}}{\partial K_{\text{acc}}} + \frac{\partial \mathcal{L}}{\partial S^\top} \cdot Q_{\text{block}}$        ▷ $[B_N \times D]$
42: $\quad$ **end for**
43:
44: $\quad \frac{\partial \mathcal{L}}{\partial K_{\text{acc}}} \leftarrow \frac{\partial \mathcal{L}}{\partial K_{\text{acc}}} \cdot \alpha$                  ▷ Apply scale factor
45: $\quad$ Store $\frac{\partial \mathcal{L}}{\partial K}[z, \mathcal{N}, h, :] \leftarrow \frac{\partial \mathcal{L}}{\partial K_{\text{acc}}}$          ▷ Write dK
46: $\quad$ Store $\frac{\partial \mathcal{L}}{\partial V}[z, \mathcal{N}, h, :] \leftarrow \frac{\partial \mathcal{L}}{\partial V_{\text{acc}}}$          ▷ Write dV
47: **end for**

---

### A.4.3 Latency Benchmarks

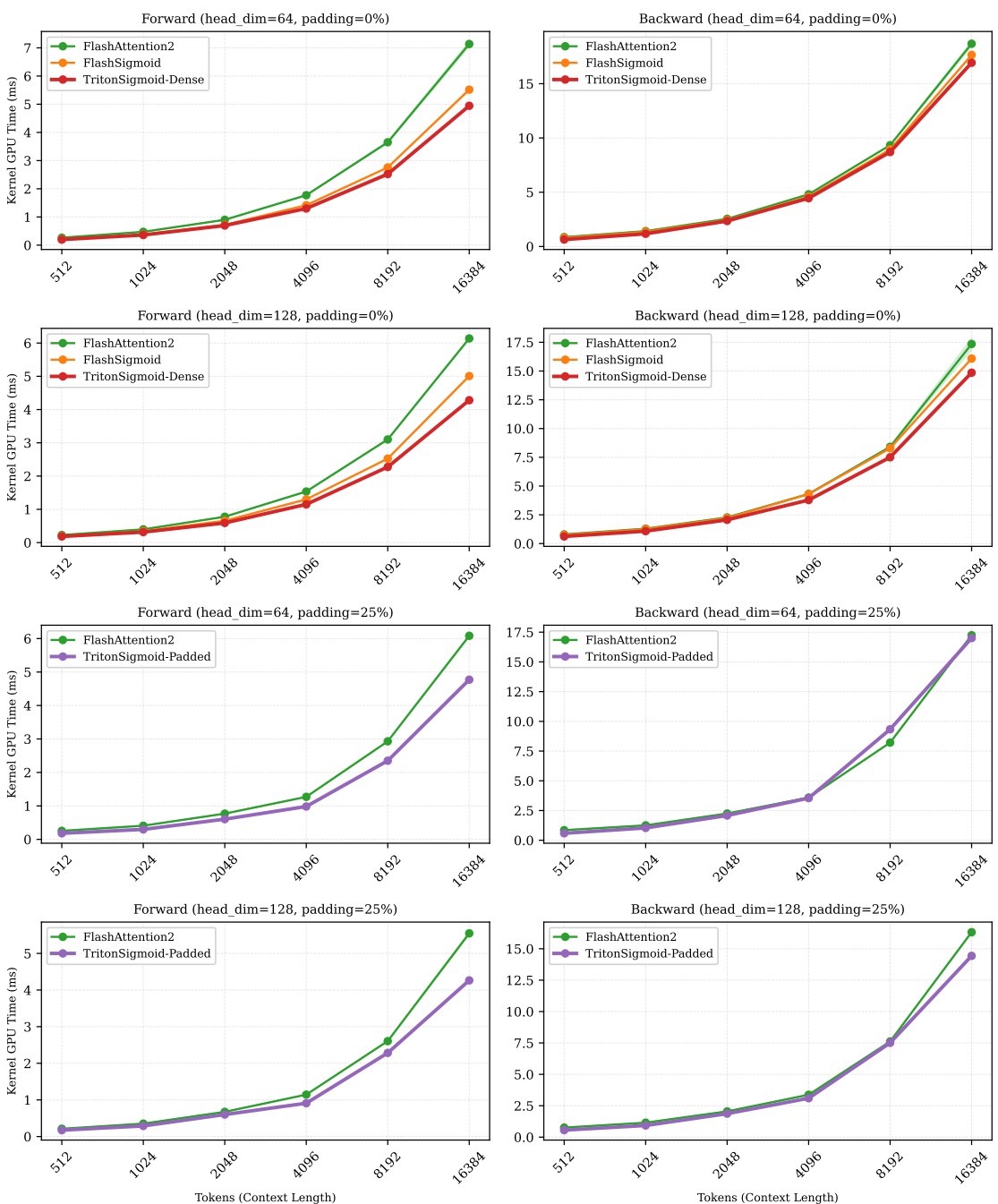

Figure 8: **Kernel execution time comparison.** GPU kernel latency (milliseconds) across head dimensions (64, 128), padding levels (0%, 25%), and forward/backward passes. Rows 1–2: no padding. Rows 3–4: 25% padding, FlashSigmoid unavailable (no padding support). Standard Attention excluded (60–85% slower than optimized kernels). TritonSigmoid achieves lowest latency in most configurations, particularly with padding. Shaded regions: 99% confidence intervals. Context lengths: 512–16,384 tokens. H100 GPU, BF16 precision.

