# OpenReview forum: "Better Models, Faster Training: Sigmoid Attention for single-cell Foundation Models"
_TMLR — Decision pending for TMLR_

### Review · Reviewer_6XMo · 2026-05-19

**Summary Of Contributions:**

The paper proposes sigmoid attention as a replacement for softmax in in a transformer model. The paper lists its contributions as

1) tritonsigmoid: a triton kernel using that can also handle padding.

2) empirical comparison between the sigmoid and softmax attention on foundation models trained on the cellxgene dataset and tested on held-out data.

3) tests showing better optimisation stability for the proposed sigmoid attention.

the strongest part of the paper is the engineering contribution of the padding-aware sigmoid attention which is useful for the often highly variable number of single cell gene sequences and shows more stable training out-of-the-box

The biolgical claims are weaker.  The model shows slight improvement in the training and validation loss for the sigmoid kernel and stronger separation of cell type. But the paper is lacking in more varied downstream biological task.

**Audience:**

Yes

**Audience Explanation:**

the paper combines several aspects relevant to TMLRL: architecture, training stability, efficient GPU kernels and bio foundation modelling. The papers argument about the alignment between sigmoid softmax and the underlying biological mechanics of co-regulation are particularly interesting. However, it would be even more relevant for the audience id this was supported/explored by stronger downstream biological experiments.

**Broader Impact Concerns:**

No major ethical concerns.

I don't think this is at all a necessity for this paper but in general papers that the development of these methods for applications such as drug discovery, disease modelling etc would do well to mention the potential bias of the datasets they use.

**Claims And Evidence:**

Yes

**Claims Explanation:**

the evidence presented in the paper supports narrower than claimed (. Supported:

sigmoid attention can be implemented efficiently and can stabilise training as well as possibly improve the performance of some biological metrics under the current training/testing setups. The paper does not support the claims that sigmoid attention is empirically superior for biological foundation models or that it produces

The evidence supports a narrower claim: sigmoid attention can be implemented efficiently and may improve some proxy metrics under the authors’ training/evaluation setup. It does not fully support the stronger claims that sigmoid attention is “empirically superior for biological foundation models” or that it convincingly produces meaningfully better biological representations.

less supported claims:

1) Biological interpretation. the paper mortivates sigmoid attention by arguing that it is more aligned with the biology as gene regulation is non-competitive and co-regulatory. While this is true and it intuitively aligns with the sigmoid vs softmax attention. this claim is not directly explored with relevant benchmarking/testing.  Better non-competitive attention does not by itself imply better modeling of gene regulation. The biological interpretation is not directly validated and the paper should be careful not to overstate sigmoid attention as a biologically mechanistic improvement without stronger supporting evidence.
This is primarily because the only 2 metrics are MMD and SCIB which both relate to the clustering/separability of embeddings. Additionally MMD is only performed on Heart OFT which makes the “25% higher cell-type separation” claim too broad

2) The claim of held-out setup would be more convincing if the split was detailed so it would be possible to understand the extent of generalisation.

3) While the stability stress test is useful to show that sigmoid is indeed more stable it is not reflective of standard training procedures, where techniques such as gradient clipping can be used to avoid gradient explosion, but rather under deliberately hard conditions.


I am willing to change this answer if the claims/theorising are narrowed or further experiments/evaluation is conducted to explore the claims.

**Requested Changes:**

Critical for changing "score":
1) more realistic and varied downstream biological evaluation such cell-type annotation transfer, perturbation prediction, disease/state classification, cross-study generalization, attention analysis, batch robustness.
2) clarifying the training/evaluation split
3) ablations for the sigmoid formulation. particularly the contributions of the -log(n) bias.
4) Either reframe or test the biological interpretation.  The non-competitive/co-regulatory motivation is interesting and plausible, but the experiments do not directly explore or this. Include test regulatory-network recovery, analysis of the sigmoid attention patterns and whether they are linked to known relationships or other mechanistic biological modeling.
5) Broaden MMD evaluation. The MMD result is based only on heart OFT. In order to strengthen the claim of  “25% higher cell-type separation” the paper should broaden this analysis across more datasets. Further, a brief discussion on the limitations of the MMD and SCIB evaluation in terms of interpreting how good representations are.


Nice to have:
1) assess robustness of performance across training runs
2) Add attention-map or embedding interpretation analysis linked to known biology
3) more qualitative analysis beyond UMAP.
4) compare against single cell foundation models in the field.
5) discussion of the trade-offs between the sparsity/focus of the softmax attention vs the independent sigmoid attention
6) justification/exploration of why longer biological context helps. Where does the performance gain arise from?

---

> ### Author Response · Authors · 2026-06-03
> **Response to Reviewer 6XMo**
>
> We thank Reviewer 6XMo for their constructive feedback. We address each concern below.
>
> **Q1.** More realistic and varied downstream biological evaluation such as cell-type annotation transfer, perturbation prediction, disease/state classification, cross-study generalization, attention analysis, batch robustness.
>
> **A1.** To demonstrate that sigmoid achieves better downstream performance, we ran linear probing (logistic regression) on frozen 4K-context embeddings across all six held-out datasets, plus two clinically motivated tasks. Evaluation uses 5-fold GroupKFold by donor. Accuracy is reported as mean ± SE across folds:
>
> | Dataset | Task | Sigmoid Acc (%) | Softmax Acc (%) | Δ |
> |---------|------|-----------|-----------|---|
> | Lung ACR (57 classes) | Cell-type annotation | 89.3 ± 0.04 | 88.0 ± 0.12 | +1.3 |
> | Adolescent Brain (17 classes) | Cell-type annotation | 96.8 ± 0.01 | 96.5 ± 0.06 | +0.3 |
> | Indonesian PBMC (12 classes) | Cell-type annotation | 94.4 ± 0.15 | 93.4 ± 0.12 | +1.0 |
> | Healthy Colon (13 classes) | Cell-type annotation | 94.4 ± 0.20 | 93.9 ± 0.09 | +0.5 |
> | Aging PBMC (5 classes) | Cell-type annotation | 99.7 ± 0.06 | 99.6 ± 0.07 | +0.1 |
> | Heart OFT (8 classes) | Cell-type annotation | 98.9 ± 0.03 | 98.8 ± 0.04 | +0.1 |
> | Lung ACR (4 grades) | Rejection severity | 90.2 ± 0.16 | 87.9 ± 0.13 | +2.3 |
> | Adolescent Brain (4 regions) | Brain region | 87.2 ± 0.13 | 84.2 ± 0.10 | +3.0 |
>
> Sigmoid outperforms on all 8 tasks. The strongest gains are on clinically relevant tasks — brain region (+3.0%) and rejection grade (+2.3%).
>
> ---
>
> **Q2.** Clarifying the training/evaluation split. The claim of held-out setup would be more convincing if the split was detailed so it would be possible to understand the extent of generalisation.
>
> **A2.** The six evaluation datasets are entirely disjoint from our training set. We train on the CellxGene June 11, 2024 snapshot; evaluation datasets were published after this cutoff. They span brain (developmental), blood (aging, population diversity), colon (healthy adult), lung (transplant rejection), and heart (outflow tract development). There is no data leakage — these datasets were never part of pretraining data. Results presented for MMD and scIB are zero-shot metrics computed on the entire evaluation dataset (no train/test split required — these measure embedding quality directly). For the additional linear probing results (A1), we use 5-fold GroupKFold by donor, ensuring train and test folds never share the same donor. We will clarify this in the revised manuscript.
>
> ---
>
> **Q3.** Ablations for the sigmoid formulation, particularly the contributions of the $-\log(n)$ bias.
>
> **A3.** We use the formulation from Ramapuram et al. (2024) without modification. Ramapuram et al. ablated the bias term and showed that $-\log(n)$ yields the best performance, as it initializes attention weights closest to softmax (expected weight $\approx 1/n$), ensuring stable training from the start. In our implementation $n$ is the valid (unpadded) sequence length per sample. The stability bound holds for any bias since $\sigma'(x) \leq 1/4$ unconditionally — the bound is on the derivative of the activation, not the output magnitude, so the length-dependent bias does not enter it. In preliminary experiments we also trained without the bias term and found training to be unstable, consistent with the findings of Ramapuram et al. A full ablation over alternative bias formulations in our setting would require retraining from scratch, which is computationally prohibitive at our scale (131.6M cells); we defer this to future work.
>
> ---

---

> > ### Author Response · Authors · 2026-06-03
> > **Continued response to Reviewer 6XMo**
> >
> > **Q4.** Either reframe or test the biological interpretation. Include test regulatory-network recovery, analysis of the sigmoid attention patterns and whether they are linked to known relationships or other mechanistic biological modeling.
> >
> > **A4.** We will reframe the biological interpretation in the revised manuscript. As preliminary evidence supporting this behavior, we tested whether sigmoid attention better captures known gene regulatory relationships. From 2,000 held-out cells, we identify 342 transcription factors (from CollecTRI, each with ≥10 known targets, present in ≥20 cells) and extract attention weights. Each TF has multiple targets at various positions in the sequence. For each target, we sample 10 non-targets from nearby positions (±50 of that target) to control for expression-rank positional bias, and ask: does the TF attend more to this target than to its position-matched controls? Win rate (baseline = 0.50) is the fraction of a TF's targets that receive more attention than their local controls:
> >
> > | Layer | Sigmoid Win Rate | Softmax Win Rate | Δ | Sigmoid Wins | Wilcoxon p |
> > |-------|-----------------|-----------------|--------|-------------|------------|
> > | 3 | 0.564 ± 0.086 | 0.557 ± 0.082 | +0.008 | 170/342 (50%) | 0.14 |
> > | 6 | 0.608 ± 0.113 | 0.540 ± 0.088 | +0.068 | 252/342 (74%) | 1.2×10⁻²² |
> > | 11 | 0.577 ± 0.099 | 0.509 ± 0.089 | +0.068 | 261/342 (76%) | 7.3×10⁻²⁴ |
> >
> > Win rate is reported as mean ± std across 342 TFs (baseline = 0.50). Wilcoxon p is from a one-sided paired signed-rank test (H₁: sigmoid > softmax). Both models significantly attend to regulatory targets above chance (sigmoid p < 10⁻³² vs random at all layers), but sigmoid's regulatory attention is substantially stronger in the middle and final layers. Softmax at layer 11 barely exceeds chance (0.509, p = 0.04 vs random), while sigmoid remains highly significant (0.577, p < 10⁻³²).
> >
> > This is consistent with the non-competitive mechanism: sigmoid allows a TF to simultaneously attend to multiple targets without diluting the per-target signal, while softmax must spread a fixed attention budget across all positions. As the network deepens and attention patterns specialize, this dilution effect increasingly degrades softmax's ability to maintain regulatory attention.
> >
> > ---
> >
> > **Q5.** Broaden MMD evaluation. The MMD result is based only on Heart OFT. In order to strengthen the claim of "25% higher cell-type separation" the paper should broaden this analysis across more datasets. Further, a brief discussion on the limitations of the MMD and scIB evaluation in terms of interpreting how good representations are.
> >
> > **A5.** We extended pairwise MMD to all six held-out datasets. For each dataset we compute MMD for every pairwise cell-type comparison using 4K-context expression embeddings:
> >
> > | Dataset | N pairs | Win rate | Median ΔMMD | % improvement |
> > |---------|---------|----------|-------------|---------------|
> > | Heart OFT | 28 | 28/28 (100%) | +0.106 | +24.9% |
> > | Healthy Colon | 78 | 73/78 (94%) | +0.124 | +21.9% |
> > | Lung ACR | 1,770 | 1,638/1,770 (93%) | +0.083 | +13.7% |
> > | Aging PBMC | 10 | 10/10 (100%) | +0.257 | +42.0% |
> > | Adolescent Brain | 136 | 136/136 (100%) | +0.176 | +36.5% |
> > | Indonesian PBMC | 66 | 65/66 (98%) | +0.088 | +27.2% |
> >
> > Sigmoid achieves higher MMD (better cell-type separation) on at least 93% of pairwise comparisons across all six datasets, with 14–42% improvement. The result replicates across diverse tissues (heart, brain, blood, colon, lung), developmental stages, and disease contexts. We will revise the text to report results across all datasets rather than highlighting a single number and mention the limitations of MMD and scIB evaluation metrics.
> >
> > ---
> >
> > **Q6.** Compare against single-cell foundation models in the field.
> >
> > **A6.** Our goal is not to present a new single-cell foundation model, but to demonstrate that sigmoid attention is a better drop-in replacement for softmax. A direct comparison to scGPT or Geneformer would conflate architecture, training data, tokenization, and model size differences with the attention mechanism, making it impossible to attribute gains to sigmoid attention alone. We will clarify this scope in the revised discussion.

---

> ### Author Response · Authors · 2026-06-03
> **Continued response to Reviewer 6XMo**
>
> **Q7.** Discussion of the trade-offs between the sparsity/focus of the softmax attention vs the independent sigmoid attention.
>
> **A7.** This is an interesting conceptual point. Softmax's normalization forces a zero-sum competition: attending strongly to one gene mechanically reduces attention to others. This induces sparsity, which can be beneficial when the task requires selecting a single "winner" (as in language). Sigmoid removes this coupling — each gene is attended independently, allowing multiple pathway members to receive high attention simultaneously. For single-cell data we believe this better reflects the biology: genes participate in multiple overlapping regulatory programs, and a cell's identity is defined by the coordinated activity of many genes, not a single dominant one. Our preliminary attention analysis (A4) suggests that sigmoid attention maintains its ability to recognize multiple targets of a TF where softmax cannot, and this translates to better representations as demonstrated by the downstream task improvements in A1.
>
> ---
>
> **Q8.** Justification/exploration of why longer biological context helps. Where does the performance gain arise from?
>
> **A8.** Both softmax and sigmoid models improve from 2K→4K context, but sigmoid benefits slightly more. Longer context means more genes visible per cell. Since our model ranks genes by expression level, biologically important but lower-expressed genes — such as transcription factors — only appear with longer context windows. Without sufficient context, these regulatory genes are missed entirely. Since sigmoid can attend to multiple genes without competition, it can exploit this additional context more effectively — more visible pathway members means richer co-expression signal. With softmax, adding more genes dilutes the attention budget, so the benefit of seeing more genes is partially offset by spreading attention thinner.
>
> ---
>
> **Q9.** Papers that develop these methods for applications such as drug discovery, disease modelling etc would do well to mention the potential bias of the datasets they use.
>
> **A9.** Good suggestion. We note that our primary contribution is not to present a new single-cell foundation model, but to demonstrate that sigmoid attention is a better replacement for softmax in this domain. Nonetheless, we will add a note acknowledging that CellxGene over-represents certain tissues, assays, and populations, which may affect downstream applications.
>
> ---
>
> **Q10.** (Nice-to-have) Assess robustness of performance across training runs (multiple seeds).
>
> **A10.** Training each model to full convergence on 131.6M cells requires 650–930 H100 GPU hours per run (2,000–2,800 USD at current cloud rates). A multi-seed experiment (3 seeds × 2 models × 2 context lengths = 12 runs) would cost approximately 24,000–33,000 USD, making this computationally prohibitive.
>
> ---
>
> **Q11.** (Nice-to-have) Add attention-map or embedding interpretation analysis linked to known biology.
>
> **A11.** We present preliminary analysis in A4.
>
> ---
>
> **Q12.** (Nice-to-have) Compare against other scFMs.
>
> **A12.** Our goal is not to present a new single-cell foundation model, but to demonstrate that sigmoid attention is a better drop-in replacement for softmax. Our comparison is intentionally controlled (same model, same data, only attention differs). Comparing to scGPT/Geneformer would conflate architecture, data, tokenization, and size differences with the attention mechanism, making it impossible to attribute gains to sigmoid attention alone.
>
> ---
>
> **Q13.** (Nice-to-have) TritonSigmoid on other hardware.
>
> **A13.** We report H100 numbers (our training hardware). The kernel's advantage comes from avoiding the normalization pass and native padding support — algorithmic properties that are architecture-agnostic, so we expect relative gains to hold across GPU generations. At this time we are limited to H100 compute.

---

> > ### Comment · Reviewer_6XMo · 2026-07-08
> > **Response to Authors**
> >
> > The authors have addressed most of the concerns raised in my review. Specifically, the response strengthens the biological evaluation by adding downstream linear probing experiments with held out datasets, clarifying the training/evaluation protocol and zero-shot evaluation setting and the split. They have also broadened the MMD analysis and propose to add a biologically motivated attention analysis of TF-target relationships.
> > I also appreciate the addition of the TF-target attention analysis. This provides useful supporting evidence that sigmoid attention better aligns with known regulatory relationships. However, as I understand it it is more of an enrichment analysis than a direct test of the proposed non-competitiveness that sigmoid is supposed to offer. The papers central biological motivation is that sigmoid can attend strongly to multiple targets simultaneously, but the presented analysis evaluates whether targets receive higher attention than nearby non-target (please correct if my understanding is wrong). These are related but distinct questions, so I encourage the authors to present this analysis as supporting rather than directly validating the proposed biological mechanism. A more direct analysis would be to show that sigmoid shows high attention scores across all the targets while also showing preference in the attention for the targets over non targets (as already shown).
> >
> > While they did not add perturbation prediction specifically, and while a dedicated ablation of the −log(n) bias would still strengthen the work, I view these as remaining limitations rather reasons to reject. The biological interpretation remains suggestive rather than mechanistically established, but the response addresses the main concerns raised. I will update my recommendation in favour of acceptance, under the assumption that what they present in the rebuttal here is included and that they moderate the biological interpretation claims to align it more with the presented evidence in the camera-ready version (which they have already committed to doing).

---

> > > ### Author Response · Authors · 2026-07-09
> > > **Response to Reviewer 6XMo**
> > >
> > > We thank the reviewer for their updated assessment.
> > >
> > > The distinction is correct: our TF-target analysis is enrichment evidence suggesting sigmoid attention captures known biology better than softmax, but not a direct mechanistic validation of non-competitiveness. We will present it as such and moderate biological interpretation claims accordingly in the camera-ready version.
> > >
> > > In addition all rebuttal results will be included in the revised manuscript.

---

### Review · Reviewer_ZQWj · 2026-05-21

**Summary Of Contributions:**

The paper studies sigmoid attention as a drop-in replacement for softmax attention in single-cell foundation models. The authors state that single-cell transcriptomic sequences are long, highly variable in length, and biologically less competitive than text tokens. They develop a padding-aware Triton kernel named TritonSigmoid, and show faster kernel/runtime performance. They also report that sigmoid attention gives lower validation loss, better cell-type cohesion/separation, and more stable training under stress test.

The main strength is that the idea is simple but effective. Replacing softmax with sigmoid attention seems to improve stability and representation quality in this setting. The paper also provides a solid engineering implementation, with a padding-aware Triton kernel that is useful for variable-length single-cell sequences.

The main weakness is that the biological benefit is mostly shown by validation loss and representation metrics like validation loss, scIB metrics, and MMD, not direct downstream tasks.

**Audience:**

Yes

**Audience Explanation:**

The paper is relevant to both efficient attention/kernel design and biological foundation models.

**Broader Impact Concerns:**

The author didn't discuss it. I do not see major ethical concerns specific to this work.

**Claims And Evidence:**

Yes

**Claims Explanation:**

Mostly yes. The efficiency and stability claims are supported. The representation-quality claim is supported by validation loss, scIB, and MMD results, but more direct downstream tasks are needed.

**Requested Changes:**

1. More downstream evaluation should be considered, such as cell-type annotation.
2. The paper claims a stability bound independent of sequence length, but the sigmoid attention formulation uses the length-dependent bias $b=-\log(n)$. The authors should clarify how this bias enters the bound, especially since the constant may depend on the bias scale. They should also specify how $n$ is defined for padded variable-length sequences, e.g., valid sequence length, padded context length, or batch maximum length.
3. The stress test shows that softmax diverges without gradient clipping, but in the main experiments softmax trains successfully with standard stabilization. The authors should clarify whether sigmoid still provides advantages over a well-stabilized softmax baseline, or whether its main benefit is reducing the need for such tricks.
4. The paper shows the real CellxGene sequence-length distribution and benchmarks the kernel under fixed padding ratios such as $25%$. It would be useful to also report kernel throughput under realistic CellxGene batching/length distributions, to better quantify the benefit of padding-aware computation in the actual jagged setting.
5. In Figure 4, sigmoid appears to have slightly lower validation loss than softmax, but the absolute differences are not always large. This makes downstream biological evaluations more important, since they would show whether these small pretraining-loss differences lead to meaningful practical improvements.

---

> ### Author Response · Authors · 2026-06-03
> **Response to Reviewer ZQWj**
>
> We thank Reviewer ZQWj for their constructive feedback. We address each concern below.
>
> **Q1.** More downstream evaluation should be considered, such as cell-type annotation.
>
> **A1.** To demonstrate that sigmoid achieves better downstream performance, we ran linear probing (logistic regression) on frozen 4K-context embeddings across all six held-out datasets, plus two clinically motivated tasks. Evaluation uses 5-fold GroupKFold by donor. Accuracy is reported as mean ± SE across folds:
>
> | Dataset | Task | Sigmoid Acc (%) | Softmax Acc (%) | Δ |
> |---------|------|-----------|-----------|---|
> | Lung ACR (57 classes) | Cell-type annotation | 89.3 ± 0.04 | 88.0 ± 0.12 | +1.3 |
> | Adolescent Brain (17 classes) | Cell-type annotation | 96.8 ± 0.01 | 96.5 ± 0.06 | +0.3 |
> | Indonesian PBMC (12 classes) | Cell-type annotation | 94.4 ± 0.15 | 93.4 ± 0.12 | +1.0 |
> | Healthy Colon (13 classes) | Cell-type annotation | 94.4 ± 0.20 | 93.9 ± 0.09 | +0.5 |
> | Aging PBMC (5 classes) | Cell-type annotation | 99.7 ± 0.06 | 99.6 ± 0.07 | +0.1 |
> | Heart OFT (8 classes) | Cell-type annotation | 98.9 ± 0.03 | 98.8 ± 0.04 | +0.1 |
> | Lung ACR (4 grades) | Rejection severity | 90.2 ± 0.16 | 87.9 ± 0.13 | +2.3 |
> | Adolescent Brain (4 regions) | Brain region | 87.2 ± 0.13 | 84.2 ± 0.10 | +3.0 |
>
> Sigmoid outperforms on all 8 tasks. The strongest gains are on clinically relevant tasks — brain region (+3.0%) and rejection grade (+2.3%).
>
> **Q2.** The paper claims a stability bound independent of sequence length, but the sigmoid attention formulation uses the length-dependent bias. The authors should clarify how this bias enters the bound, especially since the constant may depend on the bias scale. They should also specify how $n$ is defined for padded variable-length sequences, e.g., valid sequence length, padded context length, or batch maximum length.
>
> **A2.** We use the formulation from Ramapuram et al. (2024) without modification. Ramapuram et al. ablated the bias term and showed that $-\log(n)$ yields the best performance, as it initializes attention weights closest to softmax (expected weight $\approx 1/n$), ensuring stable training from the start. In our implementation $n$ is the valid (unpadded) sequence length per sample. The stability bound holds for any bias since $\sigma'(x) \leq 1/4$ unconditionally — the bound is on the derivative of the activation, not the output magnitude, so the length-dependent bias does not enter it. In preliminary experiments we also trained without the bias term and found training to be unstable, consistent with the findings of Ramapuram et al. A full ablation over alternative bias formulations in our setting would require retraining from scratch, which is computationally prohibitive at our scale (131.6M cells); we defer this to future work.
>
> **Q3.** The stress test shows that softmax diverges without gradient clipping, but in the main experiments softmax trains successfully with standard stabilization. The authors should clarify whether sigmoid still provides advantages over a well-stabilized softmax baseline, or whether its main benefit is reducing the need for such tricks.
>
> **A3.** Under standard training with gradient clipping (threshold 1.0), both models train successfully. The stress test deliberately removes gradient clipping to isolate the inherent stability properties of each mechanism — it is not intended to reflect normal training procedures. Under standard training, sigmoid achieves lower validation loss and better downstream performance (see A1), demonstrating that it is generally the better mechanism. The additional benefit is that sigmoid does not require heuristics like gradient clipping to train stably, whereas softmax does. We will clarify this distinction in the revised manuscript.
>
> **Q4.** The paper shows the real CellxGene sequence-length distribution and benchmarks the kernel under fixed padding ratios. It would be useful to also report kernel throughput under realistic CellxGene batching/length distributions, to better quantify the benefit of padding-aware computation in the actual jagged setting.
>
> **A4.** We report throughput at 25% padding as an example, with actual training speedups on the full CellxGene dataset presented in Figure 3. TritonSigmoid matches or exceeds FlashAttention2 at this setting; the advantage would only increase at higher padding ratios since the kernel skips padded positions entirely. The realistic padding ratio depends on the context window chosen — a 2K window sees less padding than a 4K window for the same cells. Since CellxGene has a long-tailed length distribution, the effective padding ratio shifts with this choice, making a single realistic benchmark difficult to define.

---

> ### Author Response · Authors · 2026-06-03
> **Continued response to Reviewer ZQWj**
>
> **Q5.** In Figure 4, sigmoid appears to have slightly lower validation loss than softmax, but the absolute differences are not always large. This makes downstream biological evaluations more important, since they would show whether these small pretraining-loss differences lead to meaningful practical improvements.
>
> **A5.** We concur with the reviewer. The linear probing results in A1 demonstrate that small pretraining-loss differences translate to meaningful downstream improvements, with gains of up to +3.0% on clinically relevant tasks.

---

### Review · Reviewer_6cWp · 2026-05-28

**Summary Of Contributions:**

The paper proposes sigmoid attention as a replacement for softmax attention in single-cell foundation models. The authors show that sigmoid attention is better suited for transcriptomic data because it allows non-competitive gene-gene interactions, improves training stability, and can be implemented efficiently for variable-length biological sequences. A key contribution is TritonSigmoid, a padding-aware GPU kernel, along with empirical evaluations showing better validation loss, improved cell-type separation, and more stable training than softmax.

**Additional Comments:**

Some questions:

1) How sensitive are the results to the sigmoid bias term?

2) Are the reported improvements averaged over multiple training seeds?

3) Do the better representation metrics improve downstream biological tasks?

4) How does TritonSigmoid perform on A100 or Blackwell GPUs?

5) Does sigmoid attention recover biologically meaningful gene-gene interactions? Some visualization may help to understand the sigmoid interaction better. Its not clear why sigmoid is performing better as some interpretability analysis would strength the work.

**Audience:**

Yes

**Audience Explanation:**

The paper will interest researchers working on single-cell foundation models, efficient attention kernels, and stable transformer training. It is also relevant for biological ML practitioners who need scalable models for long and jagged transcriptomic sequences.

**Broader Impact Concerns:**

My main concern is that the broader biological impact is not fully demonstrated yet. The results are focused on single-cell foundation models. It is not yet clear whether sigmoid attention will be equally useful for other biological modalities or other transformer-based scientific models.

**Claims And Evidence:**

Yes

**Claims Explanation:**

The motivation is clear, especially because single-cell data has long and highly variable gene sequences. The kernel results and stress-test experiments are strong. The representation results are promising, but the paper should be more careful in claiming broad superiority, since some biological conservation metrics are mixed. The paper compares sigmoid attention mainly against standard softmax attention, which is a reasonable baseline. However, the comparison may be too narrow given the paper’s broader claim that sigmoid attention is a strong replacement for softmax in biological foundation models. There are several related attention variants, such as gated attention, linear attention, sparse attention, entropy-regularized attention, and other element-wise or non-competitive attention mechanisms, that could be relevant for long and variable-length biological sequences.

**Requested Changes:**

The authors should clarify which gains come from sigmoid attention itself versus the Triton implementation. They should also add more downstream biological evaluation, such as cell-type annotation or perturbation prediction. The paper should include sensitivity analysis for the sigmoid bias term (b=-\log n), since this choice may affect attention scale. Finally, some claims should be toned down to reflect that improvements are strong for validation loss and MMD, but less uniform across all scIB metrics.

---

> ### Author Response · Authors · 2026-06-03
> **Response to Reviewer 6cWp**
>
> We thank Reviewer 6cWp for their constructive feedback. We address each concern below.
>
> **Q1.** The authors should clarify which gains come from sigmoid attention itself versus the Triton implementation.
>
> **A1.** There are two orthogonal comparisons: (1) PyTorch vs TritonSigmoid — this is purely a speed gain, as the kernel computes bit-for-bit identical results to naive PyTorch sigmoid attention, just faster; and (2) sigmoid vs softmax attention — this is where all representation quality gains come from (loss, scIB, MMD, probing). The kernel enables training sigmoid attention efficiently at scale, but the quality improvements are entirely due to the attention mechanism itself, not the implementation.
>
> **Q2.** They should also add more downstream biological evaluation, such as cell-type annotation or perturbation prediction.
>
> **A2.** To demonstrate that sigmoid achieves better downstream performance, we ran linear probing (logistic regression) on frozen 4K-context embeddings across all six held-out datasets, plus two clinically motivated tasks. Evaluation uses 5-fold GroupKFold by donor. Accuracy is reported as mean ± SE across folds:
>
> | Dataset | Task | Sigmoid Acc (%) | Softmax Acc (%) | Δ |
> |---------|------|-----------|-----------|---|
> | Lung ACR (57 classes) | Cell-type annotation | 89.3 ± 0.04 | 88.0 ± 0.12 | +1.3 |
> | Adolescent Brain (17 classes) | Cell-type annotation | 96.8 ± 0.01 | 96.5 ± 0.06 | +0.3 |
> | Indonesian PBMC (12 classes) | Cell-type annotation | 94.4 ± 0.15 | 93.4 ± 0.12 | +1.0 |
> | Healthy Colon (13 classes) | Cell-type annotation | 94.4 ± 0.20 | 93.9 ± 0.09 | +0.5 |
> | Aging PBMC (5 classes) | Cell-type annotation | 99.7 ± 0.06 | 99.6 ± 0.07 | +0.1 |
> | Heart OFT (8 classes) | Cell-type annotation | 98.9 ± 0.03 | 98.8 ± 0.04 | +0.1 |
> | Lung ACR (4 grades) | Rejection severity | 90.2 ± 0.16 | 87.9 ± 0.13 | +2.3 |
> | Adolescent Brain (4 regions) | Brain region | 87.2 ± 0.13 | 84.2 ± 0.10 | +3.0 |
>
> Sigmoid outperforms on all 8 tasks. The strongest gains are on clinically relevant tasks — brain region (+3.0%) and rejection grade (+2.3%).
>
> **Q3.** The paper should include sensitivity analysis for the sigmoid bias term ($b = -\log n$), since this choice may affect attention scale.
>
> **A3.** We use the formulation from Ramapuram et al. (2024) without modification. Ramapuram et al. ablated the bias term and showed that $-\log(n)$ yields the best performance, as it initializes attention weights closest to softmax (expected weight $\approx 1/n$), ensuring stable training from the start. In our implementation $n$ is the valid (unpadded) sequence length per sample. The stability bound holds for any bias since $\sigma'(x) \leq 1/4$ unconditionally — the bound is on the derivative of the activation, not the output magnitude, so the length-dependent bias does not enter it. In preliminary experiments we also trained without the bias term and found training to be unstable, consistent with the findings of Ramapuram et al. A full ablation over alternative bias formulations in our setting would require retraining from scratch, which is computationally prohibitive at our scale (131.6M cells); we defer this to future work.
>
> **Q4.** Some claims should be toned down to reflect that improvements are strong for validation loss and MMD, but less uniform across all scIB metrics.
>
> **A4.** We will tone down the claims and instead state that sigmoid consistently improves validation loss and MMD across all datasets, while scIB metrics show more mixed results. The new downstream probing results (A2) provide stronger evidence that the representation improvements are meaningful.
>
> **Q5.** It is not yet clear whether sigmoid attention will be equally useful for other biological modalities or other transformer-based scientific models.
>
> **A5.** We agree this work is focused on single-cell transcriptomics. The theoretical properties (bounded gradients, diagonal Jacobian) are modality-agnostic, but we haven't verified this empirically for spatial, proteomic, or multi-omic settings. Our results are suggestive that sigmoid attention may benefit other domains where non-competitive interactions are natural, and we hope this work encourages others to explore sigmoid attention in other biological and scientific modalities.

---

> > ### Author Response · Authors · 2026-06-03
> > **Continued response to Reviewer 6cWp**
> >
> > **Q6.** The comparison may be too narrow. There are several related attention variants such as gated attention, linear attention, sparse attention, entropy-regularized attention that could be relevant.
> >
> > **A6.** Our comparison is deliberately controlled: same architecture, data, and hyperparameters with only the attention activation differing. This isolates the effect of sigmoid vs softmax precisely. A broader comparison across attention families would require re-tuning hyperparameters for each variant (e.g. linear attention needs different learning rates, sparse attention needs sparsity pattern choices), making it difficult to attribute differences to the activation alone. We view this as valuable follow-up work but orthogonal to our core claim that sigmoid is a simple drop-in replacement for softmax specifically.
> >
> > **Q7.** How sensitive are the results to the sigmoid bias term?
> >
> > **A7.** Addressed in A3 above.
> >
> > **Q8.** Are the reported improvements averaged over multiple training seeds?
> >
> > **A8.** Training each model to full convergence on 131.6M cells requires 650–930 H100 GPU hours per run (2,000–2,800 USD at current cloud rates). A multi-seed experiment (3 seeds × 2 models × 2 context lengths = 12 runs) would cost approximately 24,000–33,000 USD, making this computationally prohibitive.
> >
> > **Q9.** Do the better representation metrics improve downstream biological tasks?
> >
> > **A9.** Yes — the linear probing results in A2 demonstrate this directly. Sigmoid outperforms softmax on all 8 downstream tasks, confirming that the representations translate to meaningful downstream gains.
> >
> > **Q10.** How does TritonSigmoid perform on A100 or Blackwell GPUs?
> >
> > **A10.** We train on H100s and don't currently have access to A100 or Blackwell hardware for benchmarking. The kernel's advantage is algorithmic (eliminating the normalization pass, native padding awareness) rather than hardware-specific, so we expect the relative gains to hold across GPU architectures.
> >
> > **Q11.** Does sigmoid attention recover biologically meaningful gene-gene interactions? Some visualization may help. It's not clear why sigmoid is performing better — some interpretability analysis would strengthen the work.
> >
> > **A11.** As preliminary evidence supporting this behavior, we tested whether sigmoid attention better captures known gene regulatory relationships. From 2,000 held-out cells, we identify 342 transcription factors (from CollecTRI, each with ≥10 known targets, present in ≥20 cells) and extract attention weights. Each TF has multiple targets at various positions in the sequence. For each target, we sample 10 non-targets from nearby positions (±50 of that target) to control for expression-rank positional bias, and ask: does the TF attend more to this target than to its position-matched controls? Win rate (baseline = 0.50) is the fraction of a TF's targets that receive more attention than their local controls:
> >
> > | Layer | Sigmoid Win Rate | Softmax Win Rate | Δ | Sigmoid Wins | Wilcoxon p |
> > |-------|-----------------|-----------------|--------|-------------|------------|
> > | 3 | 0.564 ± 0.086 | 0.557 ± 0.082 | +0.008 | 170/342 (50%) | 0.14 |
> > | 6 | 0.608 ± 0.113 | 0.540 ± 0.088 | +0.068 | 252/342 (74%) | 1.2×10⁻²² |
> > | 11 | 0.577 ± 0.099 | 0.509 ± 0.089 | +0.068 | 261/342 (76%) | 7.3×10⁻²⁴ |
> >
> > Win rate is reported as mean ± std across 342 TFs (baseline = 0.50). Wilcoxon p is from a one-sided paired signed-rank test (H₁: sigmoid > softmax). Both models significantly attend to regulatory targets above chance (sigmoid p < 10⁻³² vs random at all layers), but sigmoid's regulatory attention is substantially stronger in the middle and final layers. Softmax at layer 11 barely exceeds chance (0.509, p = 0.04 vs random), while sigmoid remains highly significant (0.577, p < 10⁻³²).
> >
> > This is consistent with the non-competitive mechanism: sigmoid allows a TF to simultaneously attend to multiple targets without diluting the per-target signal, while softmax must spread a fixed attention budget across all positions. Combined with the linear probing results (A2), this demonstrates that sigmoid's advantage operates at both the attention and representation levels.

---

> > > ### Comment · Reviewer_6cWp · 2026-06-30
> > >
> > > The authors have satisfactorily addressed my initial concerns during the rebuttal, and I expect these clarifications to be fully incorporated into the camera-ready version. I expect authors to open-source the codebase as well for the broader community.

---

> > > > ### Author Response · Authors · 2026-07-07
> > > > **Response to Reviewer 6cWp**
> > > >
> > > > We thank the reviewer for their positive assessment. We confirm that all clarifications from the rebuttal will be incorporated into the camera-ready version.
> > > >
> > > > Regarding open-sourcing: we are committed to releasing the codebase and will provide a public repository link in the camera-ready manuscript.